



# 1 Assessing branched tetraether lipids as tracers of soil organic carbon
# 2 transport through the Carminowe Creek catchment (southwest
# 3 England)

Jingjing Guo[1], Miriam Glendell[2], Jeroen Meersmans[3], Frédérique Kirkels[1], Jack J Middelburg[1],
Francien Peterse[1]
[1]Department of Earth Sciences, Utrecht University, 3584 CB Utrecht, the Netherlands
[2]The James Hutton Institute, Aberdeen, AB15 8QH, UK
[3]TERRA Teaching and Research Centre, Gembloux Agro-Bio Tech, University of Liege, 5030 Gembloux, Belgium
*Correspondence to*: Jingjing Guo (j.guo@uu.nl)
*In preparation for submission to: Biogeosciences*
**Abstract.** Soils represent the largest reservoir of organic carbon (OC) on land. Upon mobilization, this OC is either returned
to the atmosphere as carbon dioxide ($CO_2$), or transported and ultimately locked into (marine) sediments, where it will act as
a long-term sink of atmospheric $CO_2$. These fluxes of soil OC are, however, poorly quantified, mostly due to the lack of a
soil-specific tracer. In this study, a suite of branched glycerol dialkyl glycerol tetraethers (brGDGTs), which are membrane
lipids of soil bacteria, is tested as specific tracers for soil OC from source (soils under arable land, ley, grassland and
woodland) to sink (Lake Loe Pool sediments) considering a small catchment located in southwest England (i.e. Carminowe
Creek draining into Lake Loe Pool). The analysis of brGDGTs in catchment soils reveals that their distribution is not
significantly different across different land use types ($p > 0.05$), and thus does not allow tracing land use-specific soil
contributions to Lake Loe Pool sediments. Furthermore, the significantly higher contribution of 6-methyl brGDGT isomers
in creek sediments (isomerization ratio (IR) = 0.48 ± 0.10; mean ± s.d., standard deviation; $p < 0.05$) compared to that in
catchment soils (IR = 0.28 ± 0.11) indicates that the initial soil signal is substantially altered by brGDGT produced *in situ*.
Similarly, the riverine brGDGT signal appears to be overwritten by lacustrine brGDGTs in the lake sedimentary record,
indicated by remarkably lower Methylation of Branched Tetraethers (MBT'$_{5ME}$ = 0.46 ± 0.02 in creek bed sediment and 0.38
± 0.01 in lake core sediment; $p < 0.05$) and higher Degree of Cyclization (DC = 0.23 ± 0.02 in creek bed sediment and 0.32 ±
0.08 in lake core sediment). Thus, in this small catchment, brGDGTs do not allow us to trace soil OC transport.
Nevertheless, the downcore changes in the degree of cyclization and the abundance of isoprenoid GDGTs produced by
methanogens in the Lake Loe Pool sediment do reflect local environmental conditions over the past 100 years, and have
recorded the eutrophication history of the lake.



## 1 Introduction

Globally, around 1500–2000 Pg of carbon is stored in soils in the form of organic matter, which is about two times the amount of carbon in the atmosphere and three times the amount of carbon in vegetation (Janzen, 2004; Smith, 2008). Soil organic carbon (OC) plays an important role in the global carbon cycle, as subtle alterations in the soil OC reservoir may affect the concentration of atmospheric $CO_2$ and thus influence climate change (Davidson and Janssens, 2006). Atmospheric $CO_2$ that is fixed by plants through photosynthesis will be stored into soil OC pool, part of which will be transferred to streams and rivers. Upon fluvial discharge, soil OC is buried and locked into the marine or lacustrine sediment, where it will act as a long-term carbon sink. However, instead of a passive pipeline in the carbon cycle, rivers actually represent a dynamic channel, where part of the soil OC is respired back to the atmosphere, and another part may be stored in river bed or lake sediments before reaching the ocean ( Cole et al., 2007; Battin et al., 2009; Aufdenkampe et al., 2011). Hence, it is hard to determine the exact amount of soil OC that is transported to the ocean, as the dynamic processes that soil OC undergoes during transport, such as degradation and sequestration, are elusive. This is mostly due to the lack of a specific tracer to distinguish soil OC from the total pool of OC that is also comprised of plant-derived OC, aquatic produced OC, and fossil OC from rock erosion (Blair et al., 2004; Aufdenkampe et al., 2011).

To circumvent this problem, lipid biomarkers can be used to trace a specific part of the total OC pool in complex natural environmental systems (Brassell and Eglinton, 1986; Wakeham and Lee, 1993). For example, odd-numbered long chain *n*-alkanes derived from epicuticular plant waxes are widely used to detect the contribution of terrestrial OC to river-dominated marine sediments (Eglinton and Hamilton, 1967; Hedges et al., 1997; Fernandes and Sicre, 2000; Glendell et al., 2018). Similarly, lignin, an abundant biopolymer in vascular plants (Hedges et al., 1997), has been used to trace OC transport along the terrestrial-aquatic continuum by e.g., in the Mississippi River (Goñi et al., 1997; Bianchi et al., 2004), the Amazon Rivers (Hedges et al., 1986, 2000; Feng et al., 2016), and Arctic rivers (Feng et al., 2013). However, these biomarkers are derived from vegetation, which, although land-derived, is not fully representative of soil OC. Thus, in order to specifically trace and quantify the pool of soil OC, another biomarker is needed.

Branched glycerol dialkyl glycerol tetraethers (brGDGTs; Fig. 1) are membrane spanning tetraether lipids synthesized by heterotrophic bacteria that thrive in soils and peats all over the world (Weijers et al., 2006a, 2007a; Naafs et al., 2017a). Although the exact producers of these lipids are still unknown, after the detection of a brGDGT in an Acidobacterial culture (Sinninghe Damsté et al., 2011, 2014, 2018), it was assumed that Acidobacteria are the source organisms of brGDGTs. The occurrence and relative distribution of brGDGTs in a global set of modern surface soils showed that they can have 4 to 6 methyl groups attached to their alkyl backbone, where the degree of branching increases in soils from colder areas. Furthermore, brGDGTs respond to changes in soil pH by forming up to 2 cyclopentane moieties following internal cyclization, where a higher number of cyclopentane moieties corresponds to a higher soil pH (Weijers et al., 2007a). Initially, a combination of two indices, the Methylation of Branched Tetraethers (MBT) index and Cyclization of Branched





Tetraethers (CBT) index, was proposed as a proxy to reconstruct the mean air temperature (MAT) and pH of a soil (Weijers
et al., 2007a; Peterse et al., 2012). After the identification of novel brGDGT isomers that possess a methyl group at the α
and/or ω 6 position rather than at position 5 (Fig. 1) and the improvement of the chromatography method used for brGDGT
analysis, a modified temperature proxy, the MBT'$_{5ME}$ was developed (De Jonge et al., 2013, 2014b). Furthermore, the
relative abundance of 6-methyl brGDGT isomers, quantified as the Isomerization Ratio (IR), appeared to also relate to soil
pH (De Jonge et al., 2014a). Indeed, the analysis of brGDGTs in peat profiles and loess-paleosol sequences has resulted in
long-term continental paleotemperature records for various areas, e.g. in deglacial central China (Peterse et al., 2011) and
northeast China (Zheng et al., 2017), and western Europe during the early Eocene (Inglis et al., 2017).
These brGDGTs have not only been found in soils, but also in coastal marine sediments, where they have been used as the
terrestrial end-member in the Branched and Isoprenoid Tetraether (BIT) index that determines the relative contribution of
fluvially supplied soil organic matter to marine sediments, where the latter is represented by amounts of the isoprenoid
GDGT crenarchaeol (Hopmans et al., 2004). For example, the relative abundance of brGDGTs in a marine sediment core
from the Bay of Biscay revealed the early re-activation of European rivers after the last deglaciation (Ménot et al., 2006).
Furthermore, brGDGTs stored in continental margin sediments are assumed to represent an integrated climate signal of the
nearby land, and have been used as such to generate temperature records of deglacial tropical Africa (Weijers et al., 2007b),
and Pliocene North-Western Europe (Dearing Crampton-Flood et al., 2018).
Recently, however, brGDGTs have also been found to be produced in aquatic systems such as coastal marine areas (Peterse
et al., 2009b; Sinninghe Damsté, 2016), rivers (Zell et al., 2013, 2014a) and lakes (Sinninghe Damsté et al., 2009a; Tierney
and Russell, 2009; Loomis et al., 2011, 2014; Schoon et al., 2013; Weber et al., 2015, 2018), which complicates the
interpretation of brGDGT-based proxies. Aquatic production became apparent upon comparison of brGDGTs in Svalbard
fjord sediments and nearby soils. Whereas the brGDGT signal in the fjord sediments was dominated by compounds
containing cyclopentane moieties, soils where characterized by brGDGTs without cyclization (Peterse et al., 2009b). These
substantially different brGDGT signatures in combination with the increasing concentration of brGDGTs towards the open
ocean then pointed towards a contribution of *in situ* produced brGDGTs to the fjord sediments. Similarly, De Jonge et al.
(2014a) found that brGDGTs in suspended particulate matter (SPM) from the Yenisei River matched better with the pH of
the river water than that of the soils in the river catchment. Also, the distribution of brGDGTs in SPM remained constant,
whereas the Yenisei catchment spans a large latitudinal range with a large range in temperature, which led them to conclude
that the majority of the brGDGTs in SPM are produced in the river. Notably, these and subsequent studies proposed ways to
recognize *in situ* production of brGDGTs in aquatic environments. For example, the aquatic brGDGTs in the Yenisei River
were characterized by a relatively high contribution of 6-methyl brGDGT isomers (De Jonge et al., 2014a), whereas a high
degree of cyclization is an indicator of brGDGT production in coastal marine zones (Peterse et al., 2009b; Sinninghe



Damsté, 2016), for which Sinninghe Damsté (2016) proposed that a weighed number of rings in tetramethylated brGDGTs,
quantified as $\#rings_{tetra} > 0.7$ indicates a purely marine source of brGDGTs in continental margin sediments.
Here we test brGDGTs as tracers for soil OC in Carminowe Creek catchment, a small catchment in southwest England.
Previously, an attempt was made to follow OC transport from soil (source) to Lake Loe Pool, the final sink of this
catchment, using a combination of stable isotopes of bulk soil OC and plant leaf wax *n*-alkanes as fingerprints for the
different vegetation types present in the catchment (i.e. arable land, grassland, ley and woodland) (Glendell et al., 2018).
Although most land use types had a distinct *n*-alkane fingerprint, OC derived from arable land and temporary grassland (ley)
could not be distinguished (Glendell et al., 2018). Hence, the analysis of brGDGTs in the same samples may contribute to
tracing soil OC from different land use types during transport in Carminowe Creek. Moreover, changes in brGDGT
distributions in a 50 cm long sediment core from Loe Pool can be used to infer changes in soil OC transport dynamics in the
catchment over the past century, and potentially couple them to climate or anthropogenic activity related events in the
catchment area.
**2 Methods**
**2.1 Study site and sampling**
An overview of the study area and sampling sites is given by Glendell et al. (2018). Briefly, the Carminowe Creek catchment
is located in Cornwall in southwest England (50°14' N, 5°16' W), covers an area of around 4.8 km² and varies in elevation
from 0 to 80 m above sea level (Fig. 2). It is divided into two subcatchments ('north' and 'south'). The two streams converge
around 100 m before their joint outlet, and then flow into a natural freshwater lake Loe Pool (50 ha), which is separated from
the Atlantic Ocean by a natural shingle barrier. The mean annual temperature (MAT) and mean annual precipitation (MAP)
in this area are approximately 11 °C and 1000 mm year $^{-1}$, respectively. The land use in this studied catchment is dominated
by arable land and temporary grasslands (ley), which are under rotation. The steeper hillslopes are under permanent
grassland, and riparian woodland covers the areas near the creek. For this study, 74 surface soil samples (0–15 cm) were
collected along 14 hillslope transects, including 31 arable land sites, 14 permanent grassland sites, 24 temporary grassland
(ley) sites and 5 woodland sites (Fig. 2). Riverbed sediments were collected at three locations along each of the two
tributaries (upstream, midstream and downstream), and one more at the joint outlet. A 50 cm long sediment core was taken
in the lake, about 150 m away from the joint outlet. The lake core has been dated by the activity of Caesium-137 ($^{137}$Cs), and
it covers the last 100 years (Glendell et al., 2018).
**2.2 Bulk soil properties**
Total carbon contents were reported by Glendell et al. (2018). Soil pH was measured in this study using a pH meter in a soil
to water ratio of 1:5 (w:v) after shaking for two hours.



### 2.3 GDGT extraction and analysis

In total, 74 soil samples, 7 creek bed sediment and 25 lake core sediment samples were analysed for GDGTs. First, 5–7 g of the soils or 3–5 g of the sediments were freeze dried and homogenized, after which they were extracted three times with dichloromethane (DCM) : MeOH (9 : 1, v/v) using an accelerated solvent extractor (ASE 350, Dionex$^{TM}$) at 100 °C and 7.7 $\times 10^6$ Pa to obtain a total lipid extract (TLE). After addition of a known amount of $C_{46}$ GDGT internal standard (Huguet et al., 2006), the TLEs were dried under a $N_2$ stream, and then separated into apolar and polar fractions by passing them over an activated $Al_2O_3$ column using hexane : DCM (9 : 1, v/v) and DCM : MeOH (1 : 1, v/v) respectively. The polar fraction, which contains the GDGTs, was evaporated to dryness under a gentle $N_2$ stream. After this, the samples were prepared for further analysis by re-dissolving them in a hexane : isopropanol (99 : 1, v/v) mixture, and filtration through a 0.45 $\mu$m polytetrafluoroethylene (PTFE) filter.

The GDGTs were analysed on an Agilent 1260 Infinity ultra high performance liquid chromatography (UHPLC) coupled to an Agilent 6130 single quadrupole mass spectrometer (MS) with settings according to Hopmans et al. (2016). The GDGTs were separated over two silica Waters Acquity UPLC BEH Hilic columns (1.7 $\mu$m, 2.1 mm x 150 mm) preceded by a guard column with the same packing. GDGTs were eluted isocratically at a flow rate of 0.2 ml min$^{-1}$ using 82% A and 18% B for 25 min, followed by a linear gradient to 70% A and 30% B for 25 min, where A = hexane and B = hexane : isopropanol (9 : 1, v/v). Sample injection volumes were 10 $\mu$L. Ionization of the GDGTs was achieved by atmospheric pressure chemical ionization with the following source settings: gas temperature 200 °C, vaporizer temperature 400 °C, $N_2$ flow 6 L min$^{-1}$, capillary voltage 3500 V, nebulizer pressure 25 psi and a corona current of 5.0 $\mu$A. By scanning the [M+H]$^+$ ions (protonated mass) in selected ion monitoring (SIM) mode, the target compounds were detected at $m/z$ 1302 (GDGT-0), 1292 (crenarchaeol), 1050 (brGDGT–IIIa), 1048 (brGDGT–IIIb), 1046 (brGDGT–IIIc), 1036 (brGDGT–IIa), 1034 (brGDGT–IIb), 1032 (brGDGT–IIc), 1022 (brGDGT–Ia), 1020 (brGDGT–Ib), 1018 (brGDGT–Ic), with $m/z$ 744 for the internal standard. Quantitation was achieved by peak area integration of the [M+H]$^+$ ions in Chemstation software B.04.03.

### 2.4 GDGT proxy calculations

The roman numerals in following equations refer to the molecular structures of GDGTs in Fig.1. The ratios below were calculated based on the fractional abundances (indicated by using square brackets) of GDGTs. The BIT index was calculated according to Hopmans et al. (2004), and modified to also include 6-methyl brGDGTs:

$$BIT = \frac{[Ia]+[IIa]+[IIIa]+[IIa']+[IIIa']}{[Ia]+[IIa]+[IIIa]+[IIa']+[IIIa']+[crenarchaeol]} \tag{1}$$

The degree of methylation (MBT'$_{5ME}$) and relative abundances of tetra-, penta-, and hexamethylated brGDGTs were calculated following De Jonge et al. (2014b) and Sinninghe Damsté et al. (2016):



$$MBT'_{5Me} = \frac{[Ia]+[Ib]+[Ic]}{[Ia]+[Ib]+[Ic]+[IIa]+[IIb]+[IIc]+[IIIa]}$$ (2)
$\%tetra = \sum[tetramethylated\ brGDGTs] = [Ia] + [Ib] + [Ic]$ (3)
$\%penta = \sum[pentamethylated\ brGDGTs] = [IIa] + [IIb] + [IIc] + [IIa'] + [IIb'] + [IIc']$ (4)
$\%hexa = \sum[hexamethylated\ brGDGTs] = [IIIa] + [IIIb] + [IIIc] + [IIIa'] + [IIIb'] + [IIIc']$ (5)
Furthermore, the degree of cyclization (DC) was calculated according to Baxter et al. (2019):
$$DC = \frac{[Ib]+2*[Ic]+[IIb]+[IIb']}{[Ia]+[Ib]+[Ic]+[IIa]+[IIa']+[IIb]+[IIb']}$$ (6)
The isomerization ratio (IR) is the ratio between penta- and hexamethylated 6-methyl brGDGTs and the total amount of both
5- and 6-methyl penta- and hexamethylated brGDGTs (De Jonge et al., 2014a):
$$IR = \frac{[IIa']+[IIb']+[IIc']+[IIIa']+[IIIb']+[IIIc']}{[IIa]+[IIa']+[IIb]+[IIb']+[IIc]+[IIc']+[IIIa]+[IIIa']+[IIIb]+[IIIb']+[IIIc]+[IIIc']}$$ (7)
**2.5 Statistical analysis and data visualization**
The statistical analysis and data visualization were undertaken in R programming (version 3.5.2) (R Core Team, 2018).
Differences in the concentration of brGDGTs and brGDGT-based indices between different land use types (i.e. arable land,
grassland, ley and woodland), creek bed and lake core sediments were examined by one-way nested ANOVA under
generalized linear model (GLM) followed by post-hoc analysis (Tukey HSD (honest significant difference) test), and were
performed with package 'car', 'carData' and 'agricolae'. Differences were considered to be significant at level of $p < 0.05$.
To show how close our sample mean is close to the population mean, standard deviation is used (mean ± s.d.). To examine
whether brGDGT signatures could distinguish soil OC derived from different land use types, principal component analysis
(PCA) was performed with package 'FactoMineR' and 'factoextra'. The box plot and scatter plots were carried out with
package 'ggplot2'.
**3 Results**
**3.1 BrGDGTs in soils**
Most of the brGDGTs were present in all soils. Only brGDGT–IIIc and brGDGT–IIIc' were always below the detection limit
(peak height > 3x baseline), and brGDGT–IIc' was below the detection limit in 13 of the soils (three in arable land, four in
grassland and six in ley). The brGDGTs were dominated by pentamethylated (49.4 ± 3.0%, mean ± s.d., standard deviation),





followed by tetramethylated (39.7 ± 4.9%) and then hexamethylated brGDGTs (10.9 ± 2.6%; Table 1). The concentration of
brGDGTs ranged between 0.1 and 1.7 $\mu$g g$^{-1}$ soil, with average of 0.2 ± 0.1 $\mu$g g$^{-1}$ soil in arable land, 0.6 ± 0.4 $\mu$g g$^{-1}$ soil in
grassland, and 0.4 ± 0.3 $\mu$g g$^{-1}$ soil in ley (i.e. the temporary grassland). However, the concentration of brGDGTs in
woodland was 3.0 ± 1.0 $\mu$g g$^{-1}$ soil, which was significantly higher than that in other land use types (0.4 ± 0.3 $\mu$g g$^{-1}$ soil; $p <$
0.05; Fig. 3a). The C-normalized concentration of brGDGTs in catchment soils ranged between 2.8 to 49.8 $\mu$g g$^{-1}$ C, 8.1 ±
3.6 $\mu$g g$^{-1}$ C in arable land, 11.2 ± 6.7 $\mu$g g$^{-1}$ C in grassland, 10.5 ± 4.8 $\mu$g g$^{-1}$ C in ley, and 37.6 ± 11.0 $\mu$g g$^{-1}$ C in woodland
(Fig. 3a; Table 1). The trend of the concentration of brGDGTs along the soil transects was not obvious.
BIT index values ranged from 0.57 to 1.00 among land use types (Fig. 3b), with an average value of 0.96 ± 0.03 in
woodland, 0.90 ± 0.12 in ley, 0.88 ± 0.14 in grassland and 0.83 ± 0.09 in arable land (without significant differences, $p >$
0.05). However, the BIT values increased from hillslope to downslope along several transects in north catchment, while the
BIT values show no clear trends in south catchment (Fig. A1). The MBT'$_{5ME}$ ranged from 0.37 to 0.71 and was mostly
similar between all land use types (0.48 ± 0.04; $p >$ 0.05; Fig. 3c; Table 1). The degree of cyclization between land use types
was similar (DC = 0.23 ± 0.13; Fig. 3d; Table 1; $p >$ 0.05), likewise, the IR ranged from 0.10 to 0.60 (0.28 ± 0.01 on
average; Fig. 3e; Table 1; $p >$ 0.05), without clear trend along the soil transects. However, four transects in the north
catchment have on average significantly higher IR values (> 0.36) than the other transects in the catchment (0.24 ± 0.09; $p <$
0.05; Fig. A1). In general, the IR increases with increasing soil pH in the catchment ($r^2$ = 0.36, $p <$ 0.001).

**3.2 BrGDGTs in creek bed sediments**

All brGDGT compounds were detected in creek bed sediments, except for in the upstream site from north catchment, where
brGDGT–IIIc' was below detection limit. The brGDGTs in creek bed sediments were dominated by pentamethylated
brGDGTs (45.0 ± 0.7%), followed by tetramethylated brGDGTs (30.1 ± 4.5%), and hexamethylated brGDGTs (24.9 ±
4.7%) (Table 1). The C-normalized concentration of brGDGTs in creek bed sediments was 34.7 ± 17.4 $\mu$g g$^{-1}$ C on average
(Fig. 3a; Table 1), where the concentration increased from 32.7 $\mu$g g$^{-1}$ C to 57.0 $\mu$g g$^{-1}$ C downstream in north catchment,
and from 14.3 $\mu$g g$^{-1}$ C to 25.2 $\mu$g g$^{-1}$ C downstream in south catchment, reaching a maximum value of 59.3 $\mu$g g$^{-1}$ C at the
outlet (Fig. 5a). The concentration of brGDGTs in creek bed sediments was higher than that in soils under any land use types
except for woodland (9.6 ± 4.9 $\mu$g g$^{-1}$ C; Fig. 3a; Table 1).
The BIT values for creek sediments were on average 0.90 ± 0.06 (Fig. 3b; Table 1). The MBT'$_{5ME}$ was relatively constant
between 0.44 and 0.49, with an average of 0.46 ± 0.02. The DC ranged from 0.21 to 0.25 in the creek sediments with an
average of 0.23 ± 0.02 (Fig. 3e; Table 1). The IR was relatively invariable with an average of 0.48 ± 0.10 (Fig. 3e; Table 1).
The brGDGT-based indices in creek bed sediments were similar with the indices in soils, except for IR, which was higher
than that in soils under any land use types (0.28 ± 0.11; Fig. 3; Table 1).



### 3.3 BrGDGTs in Lake Loe Pool sediment core

All brGDGTs were detected in the lake sediment core, except at 20 cm depth, where brGDGT–IIIc' was below the detection limit. The brGDGTs in the lake sediments were mainly dominated by pentamethylated brGDGTs (50.2 ± 1.8%), followed by tetramethylated brGDGTs (28.9 ± 0.7%), and hexamethylated brGDGTs (21.0 ± 1.4%; Table 1). The amount of brGDGTs in lake core sediment ranged from 19.9 to 48.0 $\mu$g g$^{-1}$ C (Fig. 3a; Table 1). The brGDGT concentration in the surface sediment (0–2 cm), of 37.7 $\mu$g g$^{-1}$ C, which was about 1.6 times lower than that in the creek sediment at the outlet (Fig. 5a), increased to a maximum of 48.0 $\mu$g g$^{-1}$ C around 11 cm depth, and then decreased to a minimum of 19.9 $\mu$g g$^{-1}$ C at 23 cm depth (Fig. 6b). The concentration of GDGT-0 ranged between 9.0 $\mu$g g$^{-1}$ C and 27.1 $\mu$g g$^{-1}$ C with an average of 17.4 ± 6.0 $\mu$g g$^{-1}$ C, concentration of crenarchaeol ranged from 0.6 $\mu$g g$^{-1}$ C to 1.4 $\mu$g g$^{-1}$ C with an average of 1.0 ± 0.2 $\mu$g g$^{-1}$ C in the lake sediment core. In general, the concentration of brGDGTs in lake core (34.0 ± 8.7 $\mu$g g$^{-1}$ C; Table 1) was similar with that in river and in woodland, while it was significantly higher than the brGDGTs in soils except for the woodland (9.6 ± 4.9 $\mu$g g$^{-1}$ C; $p < 0.05$; Fig. 3a; Table 1).

The BIT values for the lake sediment core were rather uniform, varying between 0.95 and 0.97 (Fig.3b). Similarly, the values of MBT'$_{5ME}$ along the lake core ranged only between 0.36 to 0.39. The MBT'$_{5ME}$ of 0.37 for the lake surface sediment was significantly lower than that in creek bed sediment (0.46 ± 0.02; $p < 0.05$; Fig 3c; Fig. 5b). Conversely, the DC in the lake surface sediment was 0.39, which was significantly higher than that in creek bed sediment (0.23 ± 0.02; $p < 0.05$; Fig. 3d; Fig. 5b). The average value of DC for the lake core sediments was 0.32 ± 0.08. The DC increased from the surface to a maximum value (0.44) at around 10 cm depth, and then decreased with slight fluctuations to 0.22 at 43 cm depth (Fig. 6c). The IR was constant downcore (0.32 ± 0.01 on average; Fig. 3e; Table 1) and was significantly lower than that in creek bed sediment ($p < 0.05$; Fig. 3e).

## 4 Discussion

### 4.1 Spatial variation of brGDGT signals in catchment soils

Spatial variations in the relative distribution of brGDGTs in all catchment soils were first evaluated by performing principle component analysis (PCA) using the fractional abundances of the 13 major brGDGTs detected. The first two principal components (PCs) explain 65.2% of the variance in the dataset. PC1 describes 49.5% of the variance, and separates acyclic brGDGT–Ia and brGDGT–IIa from all the other brGDGTs (Fig. 4a). In line with this observation, PC1 has a strong positive relationship with the degree of cyclization of brGDGTs in the soils ($r^2 = 0.97$; Fig. 4c). PC2 describes another 15.7% of the variance, and separates tetramethylated brGDGTs as well as most of the 6-methyl brGDGTs from the majority of the 5-methyl penta- and hexamethylated brGDGTs. As a result, PC2 is negatively correlated with MBT'$_{5ME}$ ($r^2 = 0.49$; Fig. 4d) as well as the IR ($r^2 = 0.58$; Fig. 4e) in soils. Despite the clear relation of the first two PCs with the degree of cyclization and





the degree of methylation, respectively, the position of the soils in the PCA diagram reveals that different land use types are
largely overlapping (Fig. 4b). Indeed, the brGDGTs indices for different land use types are not significantly different ($p >$
0.05; Fig. 3), making it difficult to distinguish the provenance of soil OC solely based on brGDGT signatures.
Indeed, previous work has also shown that brGDGT distributions are not primarily affected by land use. For example,
brGDGTs in soils along an altitudinal transect in the Ethiopian highlands revealed that brGDGTs mainly reflect the decrease
in temperature with increasing elevation, regardless of drastic changes in land use along the transect (Jaeschke et al., 2018).
However, other studies report that vegetation cover does exert a great influence on brGDGT signatures in soils from
Minnesota and Ohio, USA (Weijers et al., 2011), around Lake Rotsee, Switzerland (Naeher et al., 2014), in the Tibetan
Plateau (Liang et al., 2019), and paddy and upland soils from subtropical (China and Italy) and tropical (Indonesia,
Philippines and Vietnam) areas (Mueller-Niggemann et al., 2016). The explanations for the similar distribution of brGDGTs
under different land use types in the Carminowe Creek catchment could be the rotation and ploughing in land use in
combination with the turnover time of brGDGTs. Although the soil bacterial community composition is generally different
across distinct land use types (Fierer & Jackson, 2006; Steenwerth et al., 2003), the regular rotation (generally less than 5
years) of arable land and temporary grassland (ley) in the catchment (Glendell et al., 2018) may create a mixed bacterial
community under all vegetation types. Beyond vegetation, regular ploughing as applied across the Carminowe catchment
soils (arable land and ley) is recognized to have a more dominant, long-last effect on microbial communities (Drenovsky et
al., 2010). Moreover, brGDGTs in soils have a relatively long turnover time of ca. 18 years (Weijers et al., 2010), especially
when compared to the cropland rotation time. These factors may contribute to the relatively similar brGDGT signal in all
soils in the Carminowe catchment, further limiting the variation in brGDGT signals in catchment soils.
Some spatial trends are visible in spite of the overall comparable brGDGT signals across the catchment (Fig. A1), which
may be explained by variations in other environmental factors than land use or vegetation. Mean air temperature and soil pH
have been shown to be the main factors controlling the distribution of brGDGTs in soils worldwide ( Weijers et al., 2007a;
Peterse et al., 2012; De Jonge et al., 2014b). However, in the small (ca. 4.8 km$^2$) Carminowe Creek catchment, the annual
mean air temperature is practically the same for all soils. Similarly, the range in soil pH is relatively small among different
land use types (from 5.4 ± 0.3 in woodland to 6.6 ± 0.1 in arable land; Table 1), which makes it difficult to separate brGDGT
signals based on these parameters. Additionally, the soil water content (SWC) has been shown to affect the distribution and
abundance of brGDGTs in soils, either directly by changing the microbial community, or indirectly by altering soil
temperature, soil pH, or soil oxygen content (Dirghangi et al., 2013; Menges et al., 2014; Dang et al., 2016). The SWC
positively correlates with the abundance of brGDGTs in soils from Qinghai-Tibetan Plateau (Wang et al., 2013), as well as
in soils along an aridity transect in the USA (Dirghangi et al., 2013). Moreover, the degree of methylation of 6-methylated
brGDGTs is sensitive to the SWC, especially in semi-arid and arid regions (Dang et al., 2016). Although MAP is also the
same for the whole catchment, the subtle altitudinal differences in this small creek catchment (i.e. 0-80 m above sea level)



may result in an increase in SWC from hilltop to downslope. This would introduce just enough variability in SWC to explain
some of the trends in brGDGT signals along hillslope transects. In the north of the catchment, the BIT index values gradually
increase > 0.4 from the presumably better aerated soils at the hilltops towards the wetter soils closer to the creek for two of
the transects (Fig. A1). The change in BIT index values is driven by both an increase in the amount of brGDGTs and a slight
decrease in crenarchaeol concentrations, similar to previous findings (Dirghangi et al., 2013; Wang et al., 2013; Menges et
al., 2014). The trend in BIT is likely enhanced by the (minor) change in soil pH along these two transects (from 6.2 to 6.1
and from 6.6 to 5.7), which may influence the BIT index as a result of the generally positive relation of crenarchaeol
concentrations and a negative relation of brGDGT concentrations with increasing soil pH (Weijers et al., 2006b; Peterse et
al., 2010). Nevertheless, these trends in the BIT index are visible in only two of the hilltop transects in the north part of the
catchment.
Interestingly, also the IR is significantly higher in soils along four transects in north catchment (> 0.36) compared to the
average IR value for the rest of the catchment ($0.24 \pm 0.09$; $p < 0.05$). The majority of the sites with higher IR are in
cropland except for those in the Transect-1, which is under grassland (Fig. A2). Although a relative increase in 6-methyl
brGDGTs has been linked to higher soil pH in the global soil dataset (De Jonge et al., 2014a), this relation is not so strong in
the catchment soils ($r^2 = 0.36$, p < 0.001), likely due to the relatively minor range and variation in soil pH (from $5.4 \pm 0.3$ to
$6.6 \pm 0.1$). Nevertheless, the soils with high IR values in the north catchment have pH values > 6.0 with an average value of
$6.6 \pm 0.1$.

### 4.2 Tracing brGDGTs from soils to creek bed sediments

Based on the similar brGDGT signatures for soils under different land use types, these compounds cannot be used to trace
back the exact source of the soil OC after mobilization and transport throughout the catchment. However, the concentration
and general soil signature of the brGDGTs can be compared with those in creek bed sediments to trace the transfer of OC
from the soils into the creeks. The C-normalized concentration of brGDGTs in the creek sediments is higher than that in
most of the soils ($34.7 \pm 17.4\ \mu g\ g^{-1}$ C and $9.6 \pm 4.9\ \mu g\ g^{-1}$ C respectively), except for those in the woodland soils at the
riverbanks ($37.6 \pm 11.0\ \mu g\ g^{-1}$ C; Table 1). Thus, purely based on the concentration, this suggests that brGDGTs in the creek
would be primarily derived from the woodland, which also appeared to be the main source of $n$-alkanes in creek bed
sediment (Glendell et al., 2018). However, when looking at the relative distribution of the brGDGTs, the percentage of
hexamethylated brGDGTs in creek sediments is higher than that in soils ($24.9 \pm 1.8\%$ and $10.9 \pm 0.3\%$, respectively),
whereas the percentage of tetramethylated brGDGTs is lower than in soils ($30.1 \pm 1.7\%$ and $39.7 \pm 0.6\%$, respectively; Table
1). Furthermore, brGDGTs in creek sediments have a significantly higher IR (i.e. $0.48 \pm 0.04$) than soils under any of the
land use types ($0.28 \pm 0.01$ on average in the catchment; $p < 0.05$; Fig. 3e; Table 1). This is clearly reflected in the PCA,
which separates the creek sediments from both the soils and lake sediments on PC2 that is associated with the IR (Fig. 4e).
The higher IR in the creek bed sediments can be explained by a contribution of aquatically (i.e. *in situ*) produced 6-methyl



brGDGTs. Similar contributions of 6-methyl brGDGTs, and thus higher IR, were also observed in suspended particulate
matter from the Yenisei River (De Jonge et al., 2014a), and upstream of the Iron Gates in the Danube River, where the
higher IR was coupled to in-river production facilitated by the lower flow velocity and decreased turbidity of the river water
(Freymond et al., 2017). Hence, the significantly higher IR in combination with the higher C-normalized concentrations of
brGDGTs in the Carminowe creek sediments suggests that the brGDGT signal is mainly aquatic. This implies that the initial
soil brGDGT signal is rapidly overprinted by a riverine *in situ* signal upon entering the creek. Only the IR for the
downstream site in the northern creek approaches that of the adjacent soil (IR = 0.30 and 0.38 ± 0.02, respectively; Fig. A2),
and may be explained by its use as arable land (Fig. 5a), which involves regular ploughing and subsequent soil mobilization
and implies a temporary, local overprint.
In attempt to further prove the riverine *in situ* production of brGDGTs and explain the higher IR in creek bed sediment than
that in soils, we roughly estimate the minimum amount of 6-methyl brGDGTs produced in creek to reach the higher IR. We
assume that the brGDGTs derived from woodland are completely transferred into creek without any degradation, thus, the 6-
methyl brGDGTs in creek is composed by 6-methyl brGDGTs produced *in situ* ($6'_{Me\ in\ situ}$), which we are going to estimate
the exact values, and 6-methyl brGDGTs derived from woodland ($6'_{Me\ woodland}$). The minimum amount of 6-methyl
brGDGTs produced *in situ* was calculated based on the IR equation using absolute concentration of brGDGTs (Eq. 8).
$$IR_{concentration} = \frac{6'_{Me\ creek}}{5'_{Me\ creek} + 6'_{Me\ creek}} = \frac{6'_{Me\ woodland} + 6'_{Me\ in\ situ}}{5'_{Me\ creek} + 6'_{Me\ woodland} + 6'_{Me\ in\ situ}}$$    (8)
The minimum amount of riverine *in situ* 6-methyl brGDGTs production was 10.67 $\mu$g g$^{-1}$ C, occupied 93.95% of the total
amount of 6-methyl brGDGTs in creek bed sediment we measured. Similarly, when considering the total soil transfer other
than only woodland, the minimum amount of riverine *in situ* 6-methyl brGDGTs was 10.66 $\mu$g g$^{-1}$ C, accounting for 93.86%
of the total 6-methyl brGDGTs in creek bed sediment. It proves that the riverine *in situ* production of 6-methyl brGDGTs is
thriving.
The absence of a clearly recognizable soil brGDGT signal in the creek bed sediments may be explained by the relatively
limited input of soil material into the creek. So far, river systems that have shown to transport a soil-derived brGDGT signal
are either characterized by a rainy season (e.g. the Congo River; Weijers et al., 2007b; Hemingway et al., 2017), or have
experienced a recent episode of extreme rainfall (e.g. the Danube River, >100 mm in 3 days causing a 100-year flood event;
Freymond et al., 2017). The Carminowe creek area does not have a distinct rainy season, and is further characterized by its
limited relief. Hence, the relatively minor input of soil-derived brGDGTs seems to be easily overprinted by riverine *in situ*
production. Alternatively, the soil-derived brGDGTs are preferentially degraded in an aquatic environment, resulting in a
signature that is dominated by brGDGTs that are produced *in situ*.





### 4.3 Sources of brGDGTs in the sediments of Lake Loe Pool

In theory, rivers would transport soil-derived OC together with any aquatic OC produced along the way. Once discharged, in this case into a lake, the OC would settle and then be buried into the sediments where it would act as a long-term sink of OC. However, the soil brGDGT signal cannot be recognized in the sediments from Loe Pool since it is already lost upon entering the Carminowe creek. Indeed, the PCA of the relative distributions of brGDGTs indicates that lake sediments plot completely separated from both the soils and creek sediments, mostly due to a higher relative abundance of GDGT–IIIa (Fig. 4a, b). As a result, the MBT'$_{5ME}$ is significantly lower in Loe Pool sediments ($0.38 \pm 0.00$) compared to in the creek bed sediments ($0.46 \pm 0.01$; $p < 0.05$) and soils ($0.48 \pm 0.01$; $p < 0.05$; Fig. 5b; Table 1). Furthermore, the DC is significantly higher in lake sediments than in both soil and creek bed sediments ($0.32 \pm 0.02$, $0.23 \pm 0.01$ and $0.23 \pm 0.01$, respectively; $p < 0.05$; Fig. 3d; Table 1). The distinct brGDGT signature of the lake sediments suggests that brGDGTs in the lake again are significantly altered compared to those in the soils and creek sediments. This implies that the riverine brGDGT signal is either replaced or overwritten in the lake.

Lacustrine *in situ* production of brGDGTs has been reported in other studies (Sinninghe Damsté et al., 2009a; Tierney and Russell, 2009; Loomis et al., 2011, 2014; Schoon et al., 2013; Weber et al., 2015, 2018). However, there are no generally recognized indicators (yet) to identify lacustrine brGDGT production, although several studies reported a "cold bias" while attempting to reconstruct the mean air temperature (MAT) based on brGDGTs in lake sediments using a soil-based transfer function (Tierney et al., 2010a). In a study on East African lakes, this cold bias was linked to a large *in situ* contribution of brGDGT–IIIa (Tierney et al., 2010a), similar to in Loe Pool. However, the East African lake dataset was generated using the 'old' chromatography method that does not separate 5-methyl and 6-methyl brGDGTs. A recent study that has re-analysed the East African Lake dataset indicates that the presumed contribution of GDGT–IIIa mainly consists of brGDGT–IIIa' (Russell et al., 2018), which is less prominent in lake Loe Pool. Although the identity of brGDGT-producer(s) in lakes still remain(s) elusive, a recent study from the stratified Lake Lugano (Switzerland) showed that brGDGTs are mostly produced in the lower, anoxic part of the water column rather than in the sediment (Weber et al., 2018). Furthermore, the combination of brGDGT analysis with molecular biological methods revealed that brGDGTs appeared to be produced by multiple groups of bacteria thriving under different redox regimes in this stratified lake. Specifically, brGDGT–IIIa occurred in the entire water column and continuously increased with depth, whereas brGDGT–IIIa' was mainly produced in the upper, oxygenated part of water column (Weber et al., 2018). Extrapolating the ecological niches of brGDGT production in Lake Lugano to Loe Pool we can speculate that brGDGT-IIIa, which is dominating the brGDGT signal in the Loe Pool sediments, is mostly produced in summer, when the eutrophic state of the lake may seasonally cause the anoxic conditions favourable for its (i.e. brGDGT-IIIa) production. However, our dataset does not allow to further pinpoint the time and depth of lacustrine brGDGT production, or whether brGDGTs are solely produced in the water column of Loe Pool or also in the lake sediment.



### 4.4 Reconstructing local environmental changes based on GDGTs in Loe Pool lake sediments

Downcore variations in the brGDGT distribution of Lake Pool sediments may provide information on past environmental changes in the catchment, in spite of the lacustrine *in situ* production in Lake Loe Pool. The 50 cm deep sediment core covers about the last 100 years based on [137]Cs activity (Glendell et al., 2018). The peak activity correlated with bomb testing in the 1960s was detected at 26 cm depth (Fig. 6a), which can thus be linked to 1963 (Glendell et al., 2018).

The C-normalized concentration of brGDGTs starts to increase around 23 cm, reaching a maximum concentration of 48.0 $\mu g$ $g^{-1}$ C at 11 cm depth (Fig. 6b). The increased brGDGT concentrations coincide with an increase in the degree of cyclization (Fig. 6c), which generally responds to a change in pH, where more cyclopentane moieties correspond to a higher pH (Weijers et al., 2007a; Schoon et al., 2013). According to historical records, agriculture and anthropogenic perturbations such as mining and urban pollution intensified in the 1960s (~ 26 cm depth), which increased the input of soil and nutrients into Lake Loe Pool (Coard et al., 1983), and resulted in eutrophication (i.e. blooms of cyanobacteria and algae) since at least 1986 (~ 23 cm depth) (O'Sullivan, 1992; Flory and Hawley, 1994). Earlier studies have also recognized an increased use of farmyard manures and septic tanks at this time in the nitrogen isotopic composition of the lake sediments, and have detected higher inputs of terrestrial organic material resulting from intensified farming practices and a higher erosion rate during the 1960s to 1980s based on ratios of aquatic- and terrestrial-derived plant waxes (Glendell et al., 2018). Thus, the high brGDGT concentrations and DC in the sediments likely reflect the eutrophic conditions of the lake resulting from the increased nutrient input to the lake (Coard et al., 1983). The DC has then recorded the increase in lake water pH associated with eutrophication, whereas brGDGT concentrations express increased aquatic production. Due to remediation measures taken by the local government in 1996 (~ 12 cm depth), the eutrophication has reduced over the past twenty years (Flory and Hawley, 1994; Glendell et al., 2018). The partial recovery of the lake has likely resulted in a return to lower lake water pH, as manifested in the decrease in the DC from ~ 10 cm depth upwards (Fig. 6c).

The process of eutrophication and subsequent recovery can also be recognized in the ratio between GDGT–0 and crenarchaeol, which are isoprenoidal GDGTs produced by Archaea. Crenarchaeol is produced by ammonia oxidizing Thaumarchaeota (Sinninghe Damsté et al., 2002) in aquatic environments (Schouten et al., 2000; Powers et al., 2004) and to a lesser extent also in soils (Weijers et al., 2006a), whereas GDGT–0 is a membrane lipid that occurs in all major groups of Archaea, but is indicative of methanogens and thus anaerobic conditions, with a typical ratio of GDGT-0 and crenarchaeol > 2 (Blaga et al., 2009). The ratio of GDGT-0/crenarchaeol in the sediments of Loe Pool is > 2 throughout the entire core, and ranges between 10.9 and 24.3, indicating that at least the bottom waters of the lake have been (seasonally) anoxic over the past 100 years (Fig. 6d). The ratio reaches its maximum at 16 cm depth, suggesting that eutrophic conditions and bottom water anoxia were most severe around this time. The recovery of the lake after the remediation measures is again reflected in the return to pre-1960 values (Fig. 6d).



## 5 Conclusions

In this study, brGDGTs were tested as a tracer for the transport of soil OC from different vegetation and land use types from source (soil) to sink (lake Loe Pool) in the Carminowe Creek catchment with the aim to reconstruct the provenance of the soil OC in lake Loe Pool sediments over time. Unfortunately, brGDGT signatures in the catchment soils are not distinct for land use types, indicating that other environmental parameters have a larger influence on the distribution of brGDGTs in these soils. Although temperature and precipitation can be considered equal for all soils due to the small size of the catchment, changes in BIT index values and the relative contribution of 6-methyl brGDGTs along a part of the hilltop transects indicate that soil water content (SWC) likely exerts a control on brGDGT signals, assuming that SWC increases downslope. The regular rotation of cropland in this catchment and the relative long turnover time of brGDGTs in soils could be another reason to explain the limited spatial variation in brGDGT signals.

Comparison of the soil-derived brGDGT signals to that of creek bed sediments reveals that the soil brGDGT signal is replaced by aquatically produced brGDGTs, indicated by a substantially higher fractional abundance of 6-methyl brGDGTs in the creek. Upon discharge into the lake, the creek brGDGT signal is replaced by a lacustrine *in situ* produced brGDGT signal, which is characterized by a relatively higher DC and lower MBT'$_{5ME}$, as well as a specifically high fractional abundance brGDGT-IIIa. Despite regular ploughing of the land, the absence of a profound rainy season and limited relief likely limits the degree of soil mobilization necessary to transfer the soil-derived brGDGT signal to the lake sediments in the modern system. Still, downcore variations in GDGT distributions in the sediments of Loe Pool do reflect local environmental conditions over the past 100 years. The weighed degree of cyclization of brGDGTs as well as the ratio of isoprenoidal GDGT-0 and crenarchaeol produced by Archaea match well with the historical record of lake eutrophication induced by increased nutrient input from intensified agricultural activity in the catchment during the 1960s to 1980s, and its recovery after measures taken by the owner since 1996. Our study shows that GDGTs in sedimentary archives are good recorders of past environmental and land management change, although the ability of brGDGTs to trace soil OC along a soil-aquatic continuum requires a higher degree of soil mobilization.

**Data availability**

All data are available in the Supplementary Information.

**Author Contribution**

J.M., F.K., and F.P designed the study, M.G. and J.M. collected the sample material. J.G. conducted the biomarker analysis and interpreted the data under supervision of F.P. and J.J.M, J.G. and F.P wrote the paper with input from all co-authors.



**Competing interests**
The authors declare that they have no conflict of interest.
**Acknowledgements**
This study was supported financially by NWO Veni grant #863.13.016 to Francien Peterse. Desmond Eefting and Klaas
Nierop are acknowledged for technical support.

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








**Figure 1: Molecular structures of 5-methyl and 6-methyl branched GDGTs, GDGT-0 and crenarchaeol. The 6-methyl brGDGTs are represented by apostrophe. The structures of penta- and hexamethylated brGDGTs with cyclopentane moiety(ies) IIb', IIc', IIIb', IIIc' are tentative.**







Figure 2: Map of the Carminowe Creek catchment in southwest England showing land use types, 14 soil transects (labelled T1-14), creek bed and lake core sediment sampling locations. The coloured circles and stars indicate soil samples under different land use types and creek bed sediments along the streams, respectively. Adjusted from Glendell et al. (2018).

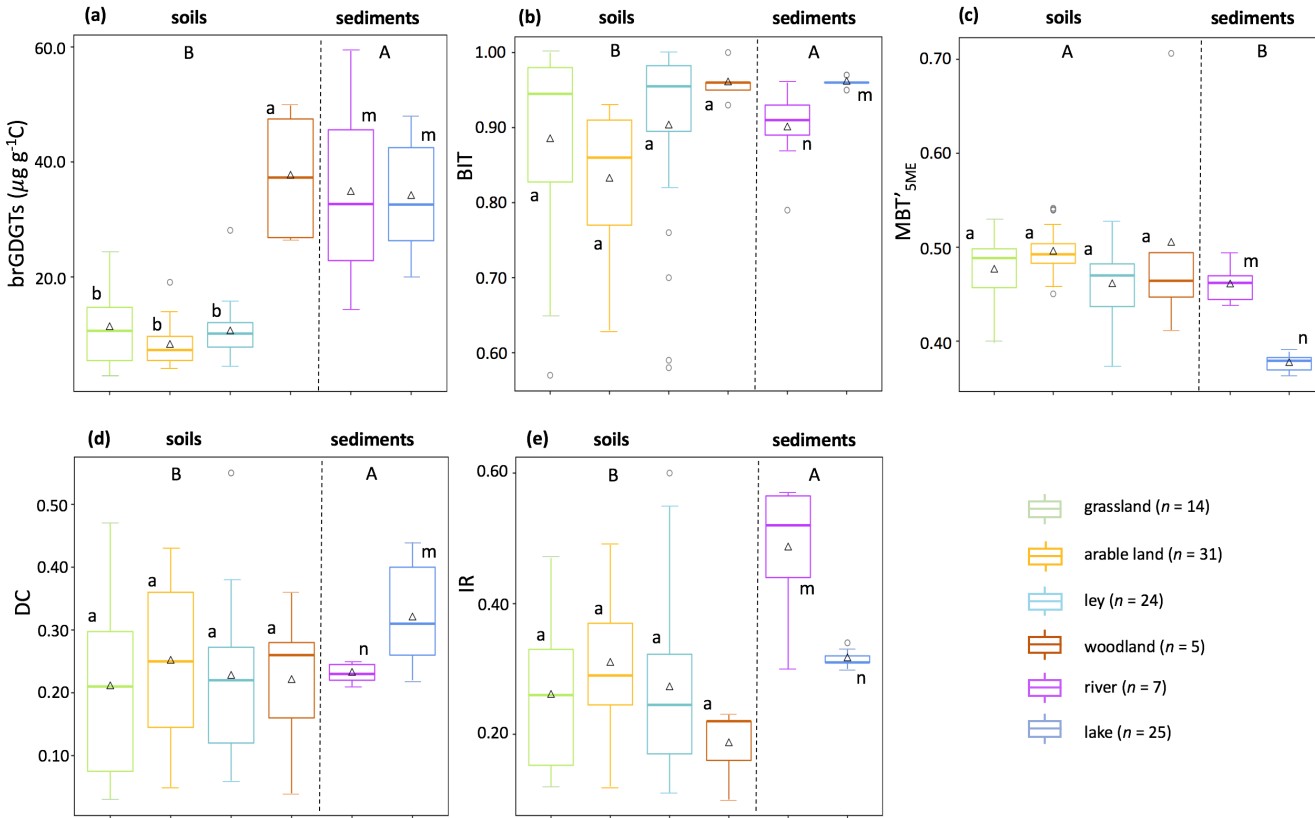

**Figure 3: Box plots displaying (a) the C-normalized concentration of brGDGTs, and brGDGT-based indices: (b) BIT index (branched and isoprenoid tetraether ratio), (c) MBT'$_{5ME}$ (methylation of 5-methyl branched tetraethers), (d) DC (degree of cyclization) and (e) IR (isomerization ratio). The triangles represent the average values, the bold line indicates the median (50th percentile), bottom and top of the box indicate first quartile (25th percentile) and third quartile (75th percentile) respectively, whiskers cover the smallest and largest value within 1.5 times of the interquartile range (i.e. the distance between the top and bottom of the box). Any data points outside the whiskers are considered as outliers. Different letters indicate differences between samples: A and B for differences between catchment soils and aquatic sediments, a and b for soils under different vegetation types, and m and n for creek bed and lake core sediments ($p < 0.05$).**



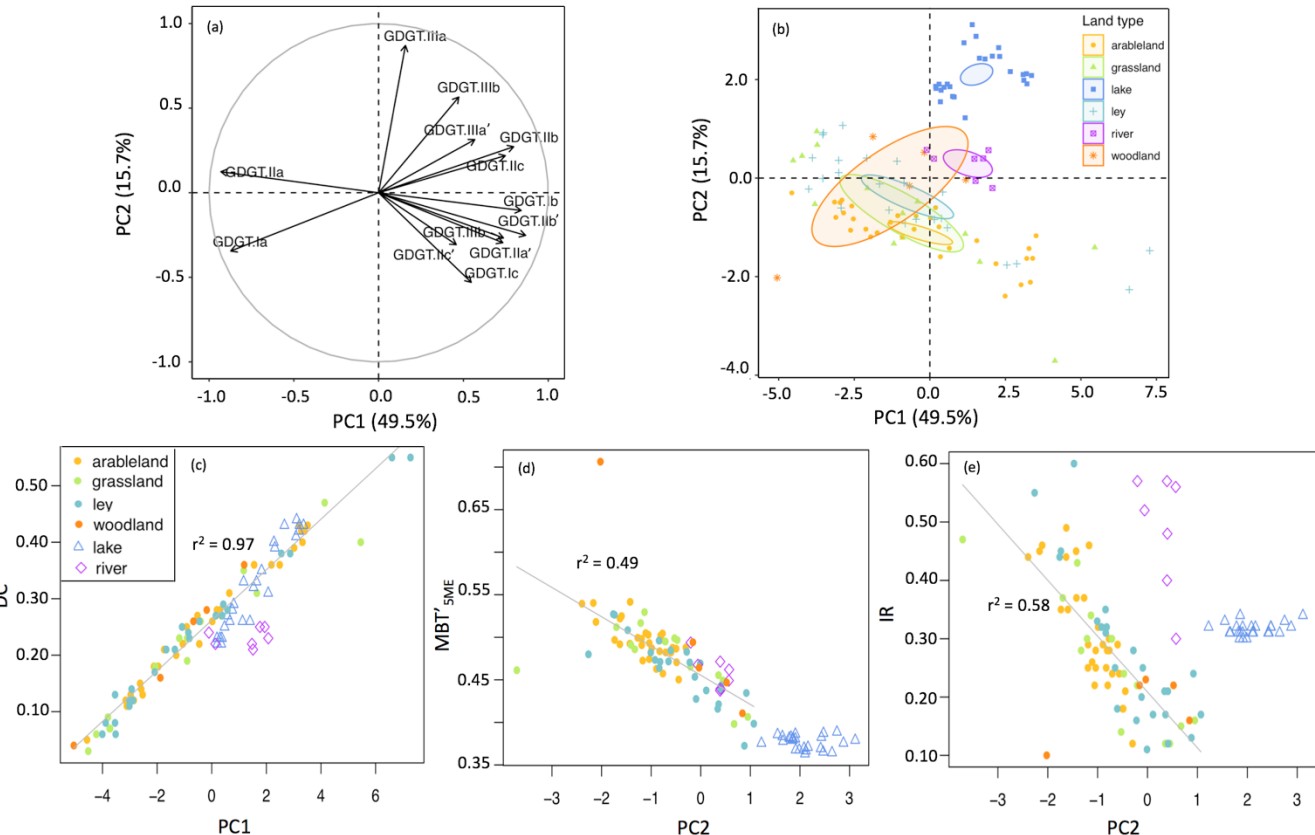

**Figure 4: PCA based on the relative abundances of 13 major brGDGTs. Figure (a) shows the distribution of 13 brGDGTs (brGDGT-IIIc and brGDGT-IIIc' are excluded as they are below the detection limit) along the first two PCs, roman numerals and English alphabet represent the compounds shown in the Fig. 2. Figure (b) shows sampling sites loading scores on the first two PCs and 95% confidence ellipses around the categories of land use: arable land ($n$ = 31), grassland ($n$ = 14), ley ($n$ = 24) and woodland ($n$ = 5), and creek ($n$ = 7) and lake ($n$ = 25). Figure (c) shows cross plots between PC1 and DC (degree of cyclization). Figure (d) and (e) show cross plots of PC2 with MBT'$_{5ME}$ (methylation of 5-methyl branched tetraethers) and IR (isomerization ratio) respectively. The linear correlation was calculated excluding creek and lake sediment.**



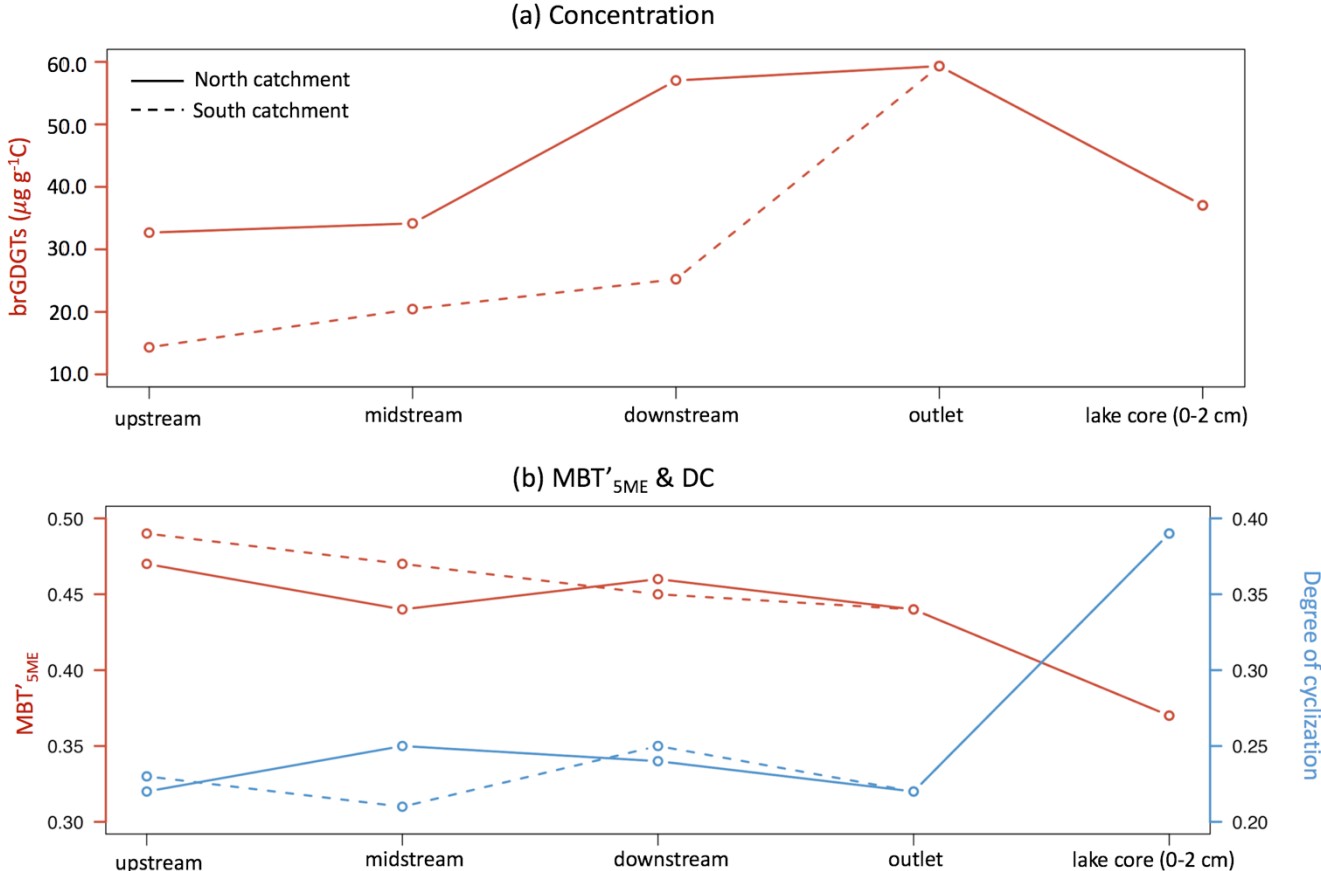

**Figure 5: Spatial variability of (a) C-normalized concentration of brGDGTs and (b) MBT'$_{5ME}$ (methylation of 5-methyl branched tetraethers) and DC (degree of cyclization) in downstream direction of both substreams in the Carminowe Creek catchment.**



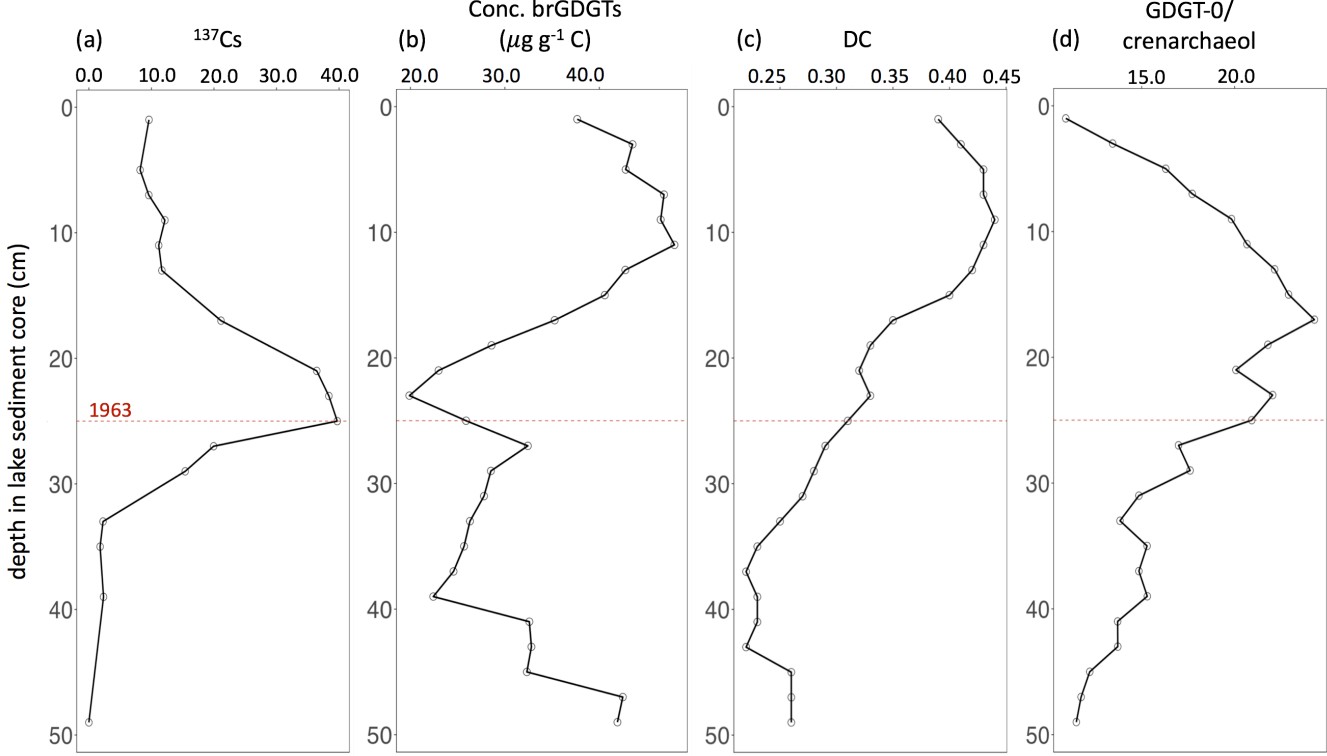

**Figure 6: Lake sediment core profiles of (a) $^{137}$Cs to date, (b) C-normalized concentration of brGDGTs, (c) DC (degree of cyclization) and (d) ratio between GDGT-0 and crenarchaeol. The red dashed line indicates the year of 1963.**





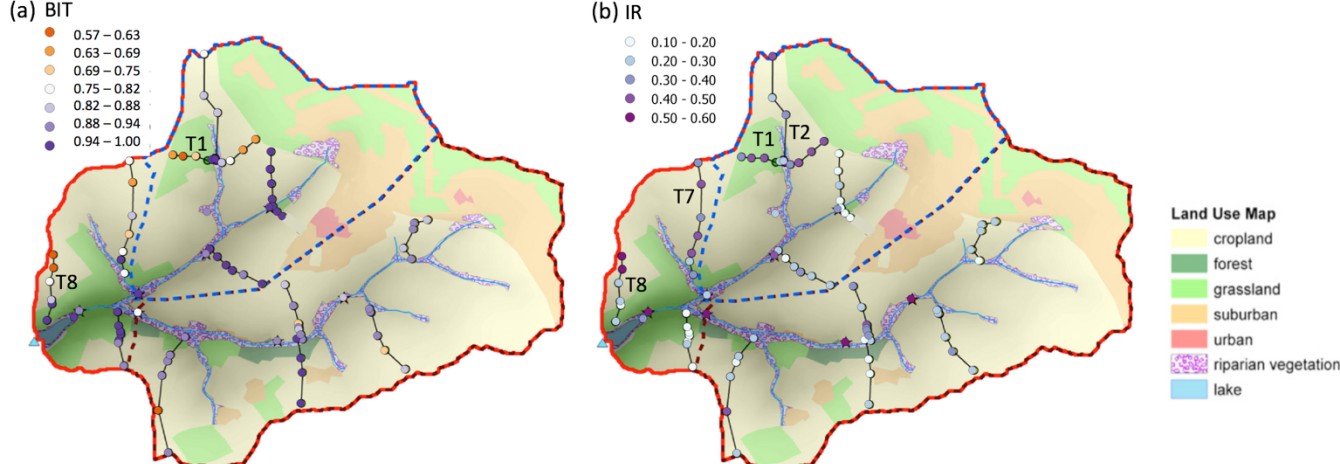

**Appendix: Spatial variability of the (a) BIT (branched and isoprenoid tetraether ratio) and (b) IR (isomerization ratio) along 14 soil transects in the Carminowe Creek catchment. The coloured circles show the concentrations and proxy values. Tx indicates soil transects discussed in the text. The background colours indicate different land use types. Adjusted from Glendell et al. (2018).**





**Table 1**. C% (carbon content), pH values, average concentrations of brGDGTs and brGDGT-based indices under different land use types. BIT (branched and isoprenoid tetraether ratio), MBT'$_{5ME}$ (methylation of 5-methyl branched tetraethers), %tetra (percentage of tetramethylated brGDGTs), %penta (percentage of pentamethylated brGDGTs), %hexa (percentage of hexamethylated brGDGTs), DC (degree of cyclization), IR (isomerization ratio) (mean $\pm$ standard deviation, s.d.).

| Land use (n) | C% * | pH | Conc. ($\mu$g g$^{-1}$ soil) | Conc. ($\mu$g g$^{-1}$ C) | BIT | MBT'$_{5ME}$ | %tetra | %penta | %hexa | DC | IR |
|---|---|---|---|---|---|---|---|---|---|---|---|
| arable (31) | 2.9 $\pm$ 0.5 | 6.6 $\pm$ 0.4 | 0.2 $\pm$ 0.1 | 8.1 $\pm$ 3.6 | 0.83 $\pm$ 0.09 | 0.50 $\pm$ 0.02 | 40.1 $\pm$ 3.1 | 49.7 $\pm$ 1.5 | 10.2 $\pm$ 1.8 | 0.25 $\pm$ 0.11 | 0.31 $\pm$ 0.10 |
| grass (14) | 5.6 $\pm$ 1.2 | 6.0 $\pm$ 0.5 | 0.6 $\pm$ 0.4 | 11.2 $\pm$ 6.7 | 0.88 $\pm$ 0.14 | 0.48 $\pm$ 0.04 | 39.8 $\pm$ 4.0 | 49.4 $\pm$ 2.3 | 10.8 $\pm$ 2.1 | 0.21 $\pm$ 0.14 | 0.26 $\pm$ 0.12 |
| ley (24) | 3.6 $\pm$ 0.9 | 6.0 $\pm$ 0.3 | 0.4 $\pm$ 0.3 | 10.5 $\pm$ 4.8 | 0.90 $\pm$ 0.12 | 0.46 $\pm$ 0.04 | 37.8 $\pm$ 3.7 | 50.2 $\pm$ 2.0 | 12.0 $\pm$ 2.9 | 0.23 $\pm$ 0.14 | 0.27 $\pm$ 0.13 |
| woodland (5) | 8.2 $\pm$ 2.1 | 5.4 $\pm$ 0.7 | 3.0 $\pm$ 1.0 | 37.6 $\pm$ 11.0 | 0.96 $\pm$ 0.03 | 0.50 $\pm$ 0.12 | 45.4 $\pm$ 13.0 | 44.4 $\pm$ 8.3 | 10.3 $\pm$ 4.8 | 0.22 $\pm$ 0.12 | 0.19 $\pm$ 0.05 |
| all soils (74) | 4.0 $\pm$ 1.8 | 6.2 $\pm$ 0.5 | 0.6 $\pm$ 0.8 | 11.5 $\pm$ 8.9 | 0.87 $\pm$ 0.12 | 0.48 $\pm$ 0.04 | 39.7 $\pm$ 4.9 | 49.4 $\pm$ 3.0 | 10.9 $\pm$ 2.6 | 0.23 $\pm$ 0.13 | 0.28 $\pm$ 0.11 |
| creek (7) | 2.3 $\pm$ 0.8 | 7.1 $\pm$ 0.2 | 0.8 $\pm$ 0.4 | 34.7 $\pm$ 17.4 | 0.90 $\pm$ 0.06 | 0.46 $\pm$ 0.02 | 30.1 $\pm$ 4.5 | 45.0 $\pm$ 0.7 | 24.9 $\pm$ 4.7 | 0.23 $\pm$ 0.02 | 0.48 $\pm$ 0.10 |
| Lake (25) | 7.5 $\pm$ 1.0 | 5.7 $\pm$ 0.2 | 2.6 $\pm$ 0.7 | 34.0 $\pm$ 8.7 | 0.96 $\pm$ 0.01 | 0.38 $\pm$ 0.01 | 28.9 $\pm$ 0.7 | 50.2 $\pm$ 1.8 | 21.0 $\pm$ 1.4 | 0.32 $\pm$ 0.08 | 0.32 $\pm$ 0.01 |

678    *From Glendell et al. (2018)