# Peer review of "Assessing branched tetraether lipids as tracers of soil organic carbon transport through the Carminowe Creek catchment (southwest England)"

_Biogeosciences, 2019_

## Referee Comment (RC1) · Anonymous Referee #3 · 28 Mar 2020

Review of "Assessing branched tetraether lipids as tracers of soil organic carbon transport through the Carminowe catchment (southwest England)" by J. Guo.  Biogeosciences Discussions, 2020.

The authors of this paper aimed to use brGDGTs as soil OC tracers in a small catchment located in southwest England and compared the concentration and distribution of these lipids in soils under different land use, riverbed and lake sediments.  They showed that the relative abundance of brGDGTs does not significantly differ between soils under different land use and that brGDGTs in the riverbed and land sediments

are mainly produced in situ (in the water column and/or sediment). Therefore, they cannot be used as soil OC tracers in this specific catchment. The analysis of brGDGTs and isoGDGTs along a lacustrine sediment core covering the last 100 yrs additionally showed that the distribution of these lipids (the degree of cyclisation of brGDGTs and the ratio of isoGDGT-0 vs. crenarchaeol) is roughly consistent with eutrophication changes over this period of time.

This study is of interest, as it is comprehensive and one of the few comparing extensively 5- and 6-methyl brGDGT distribution in soils under different land use, river and lake sediments. The paper is well-written and easy to read, and to my mind deserves publication in Biogeosciences after some revisions. The authors should sometimes be more moderate in their assertions and should avoid overinterpreting the data.

The following comments should help in improving the manuscript: Line 13: Here, the authors mention the fact some tracers are required to quantify the fluxes of soil OC. Nevertheless, brGDGTs would be more qualitative than quantitative tracers. Therefore, this sentence should be modified.

Lines 52-53: Here, I would directly say that brGDGTs are ubiquitous lipids, present in terrestrial and aquatic environments, and thus not necessarily specific soil tracers.

Line 54-55: This sentence should be rephrased, as only some of the brGDGT producers may belong to the phylum Acidobacteria. As brGDGTs were detected in various settings, it seems unlikely that they are produced by the same microorganisms everywhere.

Lines 77-93: It should be clearly mentioned somewhere that BIT index can be largely biased by in situ production of brGDGTs in aquatic settings (which was not taken into account in the initial hypothesis by Hopmans et al. 2004) and therefore should be applied with caution in coastal and lacustrine settings.

Lines 82-90: These two studies are restrictive and specific. Other examples of studies

dealing with brGDGT in situ production should be mentioned here (Miller et al., 2018, Climate of the Past; Loomis et al., 2014, GCA; Buckles et al., 2014; Biogeosciences etc.). Please also mention that in situ production of more cyclized but also more methylated brGDGTs is generally observed in aquatic vs. terrestrial settings.

Lines 100-102: In order to trace soil OC with brGDGTs, these lipids should be mainly derived from soils, with only reduced in situ production. Such an assumption should be clearly specified.

Lines 160: Were some samples analysed in replicates?

Lines 172-181: IsoGDGT-0 concentrations are only reported for the lacustrine sediments. What about the soils and the riverine sediments?

Line 227: principal component analysis instead of principle component analysis

Line 236: In Fig. 4b, a lot of samples are outside the circles (the 3 groups of soils) and do not overlap. This should be acknowledged.

Line 251: Regarding the turnover of brGDGTs in soils, please also refer to the publication by Huguet et al. (2017, GCA), with turnover times between 8 and 41 years in the same range as Weijers et al. (2010).

Lines 269-270: please specify the 2 transects along which large spatial variations in BIT are observed. T1 and T2 ? All the discussion about spatial variations in BIT and soil moisture remains very speculative. How can you explain that these variations occur only along 2 transects? What about the other transects? Are they any in situ measurements of soil moisture available to strengthen the argumentation? Or measurements in the lab (after having dried the soil samples)?

Line 277: similarly, please specify the 4 transects along which large spatial variations in IR index are observed.

Lines 281: Is the relationship between the relative abundance of 6-methyl brGDGTs

and pH given for all the soils of the catchments or only those of the 4 transects previously mentioned?

Lines 281-283: similarly, please specify to which soils correspond the different pH values (those of the 4 transects, the total dataset etc).

Lines 321-322: In addition to Congo, brGDGTs are also mainly derived from soils in other large riverine systems such as the Amazon (Kim et al., 2012, GCA) or Rhône river (Kim et al., 2015, Frontiers in Earth Science).

Line 326: why whould brGDGTs would be degraded more rapidaly in soils than in aquatic settings? This sentence should be removed as it appears too speculative.

Line 348-349: as said above, the identity of the brGDGT producers remains elusive in soils as well.

Lines 352-358: I do not see the interest of this part of this discussion on the ecological niches of brGDGTs producers in Loe Pool as it is totally speculative and has no direct link with the main aim of the paper (using brGDGTs as soil OC tracers).

Lines 359-389: in this section about local environmental changes, what about reconstruction of past temperature/pH variations with brGDGT-based indices? It would be complementary to the discussion about the lake eutrophication.

Lines 385-387: The authors should also mention the in situ production of isoGDGTs in deep lacustrine sediments, as it could bias the signal recorded in the sediments.

Lines 395-397 : I would rephrase this sentence. There is no direct evidence that soil moisture exerts a control on brGDGT distribution here and the variations in BIT were observed along 2 transects only.

Line 401: Please replace "replaced" by "mixed", as the soil brGDGT signal is not replaced by the aquatic brGDGT signal, the two signals are mixed in the sediment.

Lines 407-410: please be more moderate here, as the interpretation based on

brGDGTs is purely qualitative and complementary to previous data. I would rather say that the trends derived from GDGT data are roughly consistent with the historical record of lake eutrophication.

Lines 411: this sentence should be modified, as in the case of the Carminowe Creek catchment, this study clearly showed that brGDGTs do not record land management change and that in situ production dominates in the riverine system.

---

## Referee Comment (RC2) · R. Sparkes (Referee) · 8 Apr 2020

Guo and colleagues have carried out an in-depth study of GDGT mobilisation, transport and production in single catchment, Loe Poole Lake, in the UK. They attempt to use branched GDGTs as tracers for soil mobilisation from different parts of the catchment, but conclude that this is not possible using the available samples. Instead they use data from the creek and lake to reconstruct catchment-wide changes through the last century.

[Figure]

This is an exemplary study, demonstrating how to carry out a thorough investigation of GDGT data. The authors should be applauded for their careful application of laboratory, analytical and statistical techniques. I recommend that this paper is accepted for publication.

Minor comments are limited to typographical and grammatical changes:

Line 166: The word 'close' is repeated

Line 256: Extra space ( Weijers

Line 268 – 271: These sentences do not make it clear the direction of the trend. "BIT values gradually increase > 0.4" lacks context – do they increase or decrease with altitude? Also, be specific about whether crenarchaeol is decreasing up or down the transect

Line 277: "Interestingly, also . . ." is grammatically odd

Line 315: suggest replacing 'occupied' with 'accounted for' or similar words

—————————————————————

---

## Author Comment (AC1) · 6 May 2020

We would like to thank this reviewer for their feedback on our manuscript. Below we indicate how we will address their comments in our revised version.

Anonymous Referee #3

The following comments should help in improving the manuscript: Line 13: Here, the authors mention the fact some tracers are required to quantify the fluxes of soil OC. Nevertheless, brGDGTs would be more qualitative than quantitative tracers. Therefore,

this sentence should be modified.

Reply: We agree with the reviewer, and we will change this.

Lines 52-53: Here, I would directly say that brGDGTs are ubiquitous lipids, present in terrestrial and aquatic environments, and thus not necessarily specific soil tracers.

Reply: We chose to follow a chronological order for our introduction, and thus first introduce the discovery of brGDGTs, followed by the development of brGDGT-based proxies, additional production in different aquatic environments (i.e. coastal marine area, rivers and lakes), and the implications of mixed sources for their use as proxies. We prefer to leave this as is.

Line 54-55: This sentence should be rephrased, as only some of the brGDGT producers may belong to the phylum Acidobacteria. As brGDGTs were detected in various settings, it seems unlikely that they are produced by the same microorganisms everywhere.

Reply: We will clarify this.

Lines 77-93: It should be clearly mentioned somewhere that BIT index can be largely biased by in situ production of brGDGTs in aquatic settings (which was not taken into account in the initial hypothesis by Hopmans et al. 2004) and therefore should be applied with caution in coastal and lacustrine settings.

Reply: We will emphasize this directly after introducing aquatic brGDGT production.

Lines 82-90: These two studies are restrictive and specific. Other examples of studies dealing with brGDGT in situ production should be mentioned here (Miller et al., 2018, Climate of the Past; Loomis et al., 2014, GCA; Buckles et al., 2014; Biogeosciences etc.). Please also mention that in situ production of more cyclized but also more methylated brGDGTs is generally observed in aquatic vs. terrestrial settings.

Reply: We agree that there are many more studies that show aquatic brGDGT production than the two that are mentioned in this comment. Please note that we already listed a large number of studies on aquatic brGDGT production in lines 77-79. Our selection includes those studies that were either first in suggesting that in situ production takes place in a certain aquatic environment, provided direct evidence for in situ production, or propose (quantitative) ways to identify the aquatic contribution. We do note, however, that in situ production in lakes is not further clarified in our manuscript. One reason for this is that there is no consistent trend among lakes that enables the identification of in situ brGDGT production, in contrast to production in rivers (more 6-methyl brGDGTs) or in coastal marine environments (higher degree of cyclisation). We will add this information to the introduction of our revised manuscript and add the appropriate references.

Lines 100-102: In order to trace soil OC with brGDGTs, these lipids should be mainly derived from soils, with only reduced in situ production. Such an assumption should be clearly specified.

Reply: We will add this.

Lines 160: Were some samples analysed in replicates?

Reply: No, we did not analyze samples in replicates.

Lines 172-181: IsoGDGT-0 concentrations are only reported for the lacustrine sediments. What about the soils and the riverine sediments?

Reply: We only reported the concentration of isoprenoid GDGTs for the lacustrine sediments as we only discuss them for this environment as part of the GDGT-0/crenarchaeol ratio (section 4.4). Concentration data for GDGT-0 in the other environments will be added to the supplementary table.

Line 227: principal component analysis instead of principle component analysis

Reply: Thanks. We will correct this.

Line 236: In Fig. 4b, a lot of samples are outside the circles (the 3 groups of soils) and do not overlap. This should be acknowledged.

Reply: There are several ways to display these results. We here followed the approach of Glendell et al. (2018), who previously studied the same set of samples. The circles in Fig. 4b represent the 95% confidence interval around the mean point of the group (the enlarged symbol inside the ellipse), which is the reason why there are multiple points that plot outside the ellipse. We will clarify this in the figure caption.

Line 251: Regarding the turnover of brGDGTs in soils, please also refer to the publication by Huguet et al. (2017, GCA), with turnover times between 8 and 41 years in the same range as Weijers et al. (2010).

Reply: We will add this reference.

Lines 269-270: please specify the 2 transects along which large spatial variations in BIT are observed. T1 and T2? All the discussion about spatial variations in BIT and soil moisture remains very speculative. How can you explain that these variations occur only along 2 transects? What about the other transects? Are they any in situ measurements of soil moisture available to strengthen the argumentation? Or measurements in the lab (after having dried the soil samples)?

Reply: The BIT index values gradually increase from the hilltop downwards along Transect-1 and Transect-8. As can be seen in the table attached in supplement, Transect-1 and Transect-8 show the largest change in BIT index vales (>0.3). Transect-2, Transect-3 and Transect-7 also show an increase from hilltop downslope, albeit to a smaller degree (0.17, 0.19 and 0.04 increase, respectively). The other three transects (Transect-4, Transect-5 and Transect-6) in north catchment have stable BIT values, and the BIT values in south catchment do not show an obvious trend at all. Also based on the comments of Dr. Sparkes, we will clarify our discussion on the BIT index in a revised version. Unfortunately, the soil water content was not analyzed.

Line 277: similarly, please specify the 4 transects along which large spatial variations in IR index are observed.

Reply: We will add the specifications.

Lines 281: Is the relationship between the relative abundance of 6-methyl brGDGTs and pH given for all the soils of the catchments or only those of the 4 transects previously mentioned?

Reply: The reported relationship between the relative abundance of 6-methyl brGDGTs and pH is for all the soils in the study catchment. We will specify this in the manuscript.

Lines 281-283: similarly, please specify to which soils correspond the different pH values (those of the 4 transects, the total dataset etc).

Reply: We will further specify this.

Lines 321-322: In addition to Congo, brGDGTs are also mainly derived from soils in other large riverine systems such as the Amazon (Kim et al., 2012, GCA) or RhoÌĆne river (Kim et al., 2015, Frontiers in Earth Science).

Reply: We will add these studies.

Line 326: why would brGDGTs would be degraded more rapidly in soils than in aquatic settings? This sentence should be removed as it appears too speculative.

Reply: The line that the reviewer refers to is on purpose phrased as a potential explanation for our results, and thus meant to be speculative Note that we do not compare brGDGT degradation in soils vs an aquatic setting, but the degradation of soil-derived vs aquatic brGDGTs in the same aquatic environment. One process that could explain this process is priming. We will add this explanation and appropriate references (e.g. Bianchi, 2011) to the revised version.

Line 348-349: as said above, the identity of the brGDGT producers remains elusive in soils as well.

Reply: Yes, we agree with the referee, although so far there are more clues on the producer(s) of brGDGTs in soils than there are for aquatic systems.

Lines 352-358: I do not see the interest of this part of this discussion on the ecological niches of brGDGTs producers in Loe Pool as it is totally speculative and has no direct link with the main aim of the paper (using brGDGTs as soil OC tracers).

Reply: In this section it becomes clear that the brGDGTs in the lake sediment are not derived from soils, but are most likely produced in the lake itself. Since we can, therefore, not use the brGDGTs as tracer for soil OC, we instead use this section to further explore the environmental significance of their signature stored in the lake sediments. For this, it is important to understand the depth and season of brGDGT production in Lake Loe Pool, for which we compare our dataset with the latest insights on brGDGT production in lakes in general, i.e. the ecological niches identified in Lake Lugano (Weber et al., 2018).

Lines 359-389: in this section about local environmental changes, what about reconstruction of past temperature/pH variations with brGDGT-based indices? It would be complementary to the discussion about the lake eutrophication.

Reply: We agree with the reviewer that records of past temperature and pH variations would be a valuable addition to the discussion. However, the aquatic source of the brGDGTs in the lake sediments disqualifies the use of the transfer functions from e.g. De Jonge et al., 2014 or Naafs et al., 2017, that are based on soils. We did apply the transfer functions in the latest lake calibration (Russell et al., 2018), however, the calibration dataset only includes lake sediments from tropical east Africa, and results in reconstructed temperatures that are too high (13.7 $\pm$ 0.1 °C vs the locally historical recorded temperature of 10.9 $\pm$ 0.6 °C (average of 1978 to 2018, UK metoffice)). It thus seems that both the global soil calibration and the tropical lake calibration are not appropriate for the brGDGTs in this temperate lake, and therefore decided to not include these records in our manuscript.

Lines 385-387: The authors should also mention the in situ production of isoGDGTs in deep lacustrine sediments, as it could bias the signal recorded in the sediments.

Reply: We will add that isoGDGTs may potentially be produced in deeper sediments, although we are not aware of a study that has shown this and we can add as a reference. Given the resemblance of the trends in GDGT proxies with that of the eutrophication history of the lake, we also assume that the contribution of a deep-sediment-producer will be minor.

Lines 395-397: I would rephrase this sentence. There is no direct evidence that soil moisture exerts a control on brGDGT distribution here and the variations in BIT were observed along 2 transects only.

Reply: As we mentioned above, the trend in BIT values is evident in five out of eight transects in the north catchment, although the increase is relatively small in three of them. Based on the influence of soil water content reported in the literature (e.g. Dirghangi et al., 2013; Menges et al., 2014), and the supposedly lower ground water table at the hilltop compared to the soils downslope, we will leave this interpretation as is.

Line 401: Please replace "replaced" by "mixed", as the soil brGDGT signal is not replaced by the aquatic brGDGT signal, the two signals are mixed in the sediment.

Reply: We will change this sentence.

Lines 407-410: please be more moderate here, as the interpretation based on brGDGTs is purely qualitative and complementary to previous data. I would rather say that the trends derived from GDGT data are roughly consistent with the historical record of lake eutrophication.

Reply: We will change this accordingly.

Lines 411: this sentence should be modified, as in the case of the Carminowe Creek catchment, this study clearly showed that brGDGTs do not record land management
change and that in situ production dominates in the riverine system.

Reply: Note that we here refer to GDGTs in general, not just the brGDGTs. The land management that we mention refers to the increased use of manure and septic tanks and intensified agriculture that caused the eutrophication of the lake, and the subsequent restoration efforts that are reflected in the GDGT proxy records from the lake core (Fig. 6). The conclusion that brGDGTs in the lake sediments are produced within the lake is already clearly mentioned in line 402-403.

References: Bianchi, T. S.: The role of terrestrially derived organic carbon in the coastal ocean: A changing paradigm and the priming effect, Proc. Natl. Acad. Sci., 108(49), 19473–19481, doi:10.1073/pnas.1017982108, 2011. Dirghangi, S. S., Pagani, M., Hren, M. T. and Tipple, B. J.: Distribution of glycerol dialkyl glycerol tetraethers in soils from two environmental transects in the USA, Org. Geochem., 59, 49–60, doi:10.1016/j.orggeochem.2013.03.009, 2013. De Jonge, C., Hopmans, E. C., Zell, C. I., Kim, J.-H., Schouten, S. and Sinninghe Damsté, J. S.: Occurrence and abundance of 6-methyl branched glycerol dialkyl glycerol tetraethers in soils: Implications for palaeoclimate reconstruction, Geochim. Cosmochim. Acta, 141, 97–112, doi:10.1016/j.gca.2014.06.013, 2014. Menges, J., Huguet, C., Alcañiz, J. M., Fietz, S., Sachse, D. and Rosell-Melé, A.: Influence of water availability in the distributions of branched glycerol dialkyl glycerol tetraether in soils of the Iberian Peninsula, Biogeosciences, 11(10), 2571–2581, doi:10.5194/bg-11-2571-2014, 2014. Naafs, B. D. A., Gallego-Sala, A. V., Inglis, G. N. and Pancost, R. D.: Refining the global branched glycerol dialkyl glycerol tetraether (brGDGT) soil temperature calibration, Org. Geochem., 106, 48–56, doi:10.1016/j.orggeochem.2017.01.009, 2017. Russell, J. M., Hopmans, E. C., Loomis, S. E., Liang, J. and Sinninghe Damsté, J. S.: Distributions of 5- and 6-methyl branched glycerol dialkyl glycerol tetraethers (brGDGTs) in East African lake sediment: Effects of temperature, pH, and new lacustrine paleotemperature calibrations, Org. Geochem., 117, 56–69, doi:10.1016/j.orggeochem.2017.12.003, 2018. Weber, Y., Sinninghe Damsté, J. S.,

Zopfi, J., De Jonge, C., Gilli, A., Schubert, C. J., Lepori, F., Lehmann, M. F. and Niemann, H.: Redox-dependent niche differentiation provides evidence for multiple bacterial sources of glycerol tetraether lipids in lakes, Proc. Natl. Acad. Sci., 115(43), 10926–10931, doi:10.1073/pnas.1805186115, 2018.

Please also note the supplement to this comment:
https://www.biogeosciences-discuss.net/bg-2019-500/bg-2019-500-AC1-supplement.pdf

———————————————————

[Figure]

**Supplement:**

Table1. BIT values along 14 transects (Tx indicates the transect number, and Sx indicates the sample point, where 1 represents the hilltop and subsequent numbers are further downslope).

| | BIT | North catchment | | | | | | | | South catchment | | | | | |
|---|---|---|---|---|---|---|---|---|---|---|---|---|---|---|---|
| | | **T1** | *T2* | *T3* | T4 | T5 | T6 | *T7* | **T8** | T9 | T10 | T11 | T12 | T13 | T14 |
| hilltop | S1 | **0.65** | *0.66* | *0.77* | 0.97 | 0.99 | 0.97 | *0.77* | **0.58** | 0.92 | 0.92 | 0.84 | 0.95 | 0.84 | 0.86 |
| | S2 | **0.57** | *0.66* | *0.86* | 0.99 | 0.99 | 0.93 | *0.65* | **0.59** | 0.91 | 0.63 | - | - | 0.73 | 0.88 |
| | S3 | **0.73** | *0.77* | *0.87* | 1.00 | 1.00 | 0.94 | *0.82* | **0.80** | 0.98 | 0.90 | 0.92 | 0.98 | 0.91 | 0.88 |
| | S4 | **0.88** | *0.83* | *0.96* | 0.99 | - | 0.96 | *0.70* | **0.85** | 1.00 | 0.92 | 0.72 | 0.98 | 0.92 | 0.86 |
| | S5 | **0.95** | - | - | - | - | 0.97 | *0.76* | **0.98** | 1.00 | 0.91 | 0.91 | 1.00 | 0.91 | 0.93 |
| | S6 | - | - | - | - | - | 0.96 | *0.94* | **0.97** | 1.00 | 0.93 | 0.85 | - | 0.90 | - |
| | S7 | - | - | - | - | - | - | *0.81* | - | 0.95 | - | 0.92 | - | - | - |
| downslope | S8 | - | - | - | - | - | - | - | - | - | - | 0.92 | - | - | - |

---

## Author Response (AR1)

Dear editor,

Please find enclosed our revised manuscript titled Assessing branched tetraether lipids as tracers of soil organic carbon transport through the Carminowe Creek catchment (southwest England). We thank the reviewers for their comments and we have followed most of their suggestions, as you can read in the replies posted in the online forum and enclosed here. Below we list the major changes that we have made to the manuscript. All these changes are highlighted in our revised manuscript enclosed herewith. We hope that you find this revised version suitable for publication in Biogeosciences.

On behalf of all co-authors,
Jingjing Guo

- We have followed most of the suggestions for textual changes throughout the revised manuscript.

- In the introduction, we have elaborated on the description of the elusive producer of brGDGTs, and explained the implications for the interpretation of brGDGT-based proxies. We have also clarified and expanded the explanation of *in situ* production of brGDGTs in aquatic environments and added appropriate references.

- In discussion part, we have specified the changes in the BIT index and the IR along the soil transects, and marked them in the appendix figure. Furthermore, a table with original data of the BIT index was added into the appendix. Finally, we have added the references that were suggested by the reviewers, specifically on the river transport of brGDGTs (Kim et al., 2012; Kim et al., 2015) and the turnover time of terrestrial brGDGTs (Huguet et al., 2017).

- Figures: the color of box plot (Figure 3) was changed to make different land types easier to distinguish. The mean point of different land types in PCA plot (Figure 4b) was added.

- Additionally, we discovered that we accidentally reported the incorrect IR value in section 4.2, where we use this ratio to calculate the amount of river *in situ* brGDGT production. We have corrected this value in the revised version. Importantly, this has no influence on the interpretation of the data and our conclusions.

[revised manuscript text omitted]

Dear editor,

We are pleased to hear that our manuscript is accepted for public review. We hereby upload our final manuscript for publication on the online forum of Biogeosciences Discussions. In this version, we have incorporated the feedback provided by one of the initial referees, as indicated below. Please see our replies in italics.

On behalf of all co-authors,

Jingjing Guo

Reviewer #2

In this work Guo et al. conduct a thorough analysis on brGDGT distribution in the Carminowe Creek catchment to assess its application for tracing soil organic matter transportation. Data is well presented with comprehensive discussion. However, in the main text there is substantial amount of discussion about the application of GDGT records for past deposition environment which is not related to the title specified objective of tracing soil OM transportation. Please consider reorganizing the text or modifying the title to reflect the overall objectives of work.

*Reply: As the title indicates, the primary focus of our manuscript is indeed on tracing soil OC, and not the application of GDGTs as proxies in downcore records. In the current manuscript, the application of GDGT proxies in the paleo-domain are only limited to a few examples to indicate how GDGTs can be used once stored in a sedimentary archive. Since the GDGT signature may be altered during mobilisation, transport, and deposition, this has implications for the interpretation of paleorecords based on their occurrence in sediments. Hence, we believe this makes mentioning their function as paleo-environmental proxies worthwhile. At this stage we choose to keep the title of our work and the content of the introduction as is.*

Specific Comments:

Line 40-42, are there any numbers of estimated proportion of each OC pool from the literatures? That will show how significant is the soil OC to the total carbon pool.

*Reply: We assume that with 'total carbon pool' the reviewer means the proportion of plant-derived, aquatic produced, and fossil OC from rock erosion that together make up the total pool of OC transported by rivers. The contributions of each of those separate OC pools will depend on the catchment area and the processes that take place during transport. For the Carminowe Creek catchment that we studied, these data are not available. In part this is due to the lack of specific tracers for each of those OC pools, which is one of the motivations for our study.*

Line 49-50, Is soil OC mainly composed of humus from plants? Living soil microbe contributes very small amount of OC. This is not a strong argument why we need another biomarker for tracing soil OC.

*Reply: Soil OC has a mixture of sources, among which plant material and microbes. Even though plant material may contribute more to the soil OC pool than the microbial community, plant biomarkers can also be transported directly from their source-plant by wind, thereby bypassing the soil. Hence, plant biomarkers in any system will represent a variety of transport pathways. In contrast, brGDGTs are derived from bacterial that thrive in soils, and as such have a clear soil origin, despite their relatively minor contribution to the total soil OC pool.*

Line 57, what does the 'internal cyclisation' mean?

*Reply: internal cyclisation is the process that describes the formation of cyclopentane moieties in the alkyl backbone of the branched GDGT (Weijers et al., 2006).*

Line 98-99, is this a conclusion from Glendell et al., 2018?

*Reply: Yes, this is correct. We have added Glendell et al. as a reference at the end of this sentence.*

Line 99-100, what is the reasoning behind this speculation? What makes you to expect that brGDGTs could reflect different land types?

*Reply: The brGDGTs know a large structural diversity, which relates to several environmental parameters (pH, temperature, soil moisture availability). These parameters may also be influenced by different vegetation types, which will consequently affect the brGDGT signal.*

Line 110, in this studied catchment…

*Reply: We have changed this.*

Line 126-127, please specify column length and diameter.

*Reply: We have used about 3cm of activated $Al_2O_3$ column in a Pasteur pipette according to the general practice in Organic Geochemistry labs worldwide.*

Line 133, delete silica.

*Reply: The silica refers to the packing of the HPLC columns that we have used and is crucial information as it determines the elution behaviour of the GDGTs. We have left this sentence as is.*

Line 178, what is C-normalized concentration? Total organic carbon?

*Reply: This refers to total carbon (inorganic and organic).*

Line 195, brGDGTs concentrations of soil and lake sediment are normalize to both sediment/soil and total C, but there is only C-normalized for creek sediment, why?

*Reply: We focus on C-normalized brGDGT concentrations to enable a fair coparison between the three environments (soil, river, lake), as the mineral fraction, and thus the weight, of the samples is highly variable and would bias sediment weight-based brGDGT concentrations. We only include weight-normalized brGDGT concentrations for the soils in order to compare our data with those from the literature.*

Line 248, is this mixed bacterial community the brGDGTs producing bacteria? The assumption of 'mixed bacterial community' does not sound convincing to me. It should be the physical and chemical parameters, such as temperature, pH and redox, that directly affect the brGDGTs synthesis. These parameters may or may not change with land use or vegetation types. It is the same logic for the discussion of soil water content in the following paragraph. SWC does not directly influence the brGDGTs production, but the oxygen/$CO_2$ content that determined by SWC do.

*Reply: As of yet it is not known which bacteria produce brGDGTs, although brGDGT-Ia has been found in two species of Acidobacteria (Sinninghe Damsté et al., 2011). Acidobacteria have also been found to produce brGDGT precursur lipids (Sinninghe Damsté et al., 2014, 2018), and are currently thought of the most likely producers.*

*As a result of the orphan status of the brGDGTs, it is also unknown what drives the changes in the molecular structure. There is ongoing debate whether changes are a result of membrane adaptation as a response to changing environmental parameters (as suggested by the reviwer), or of a change in the microbial community (which repond to the same environmental parameters), where different species produce distinct brGDGTs.*

*We realize that this information had not made it to the introduction, so we have added it to the submitted version.*

Line 290-291, It will be better to just speculate based on the n-alkanes data that woodland could be a source of brGDGTs in creek sediment. Concentration can not indicate source.

*Reply: We are not so sure what the reviewer means here, as n-alkanes and brGDGTs have different sources (vegetation vs soil bacterial), and can thus not be directly compared.*

Line 302-303, totally agree, further evidence is the relatively higher pH of creek. As cited in Line 280 that IR is positively correlated to pH (De Jonge et al., 2014a).

*Reply: Thanks.*

Line 344, linked to a large in situ contribution of…

*Reply: We have changed this.*

Line 384-388, if this is the case, then you can also calculate the Methane Index (Zhang et al., 2011) to indicate anoxic lake history.

*Reply: The Methane Index is most commonly used in marine environments to assess the potential influence of methanotrophic archaea on the $TEX_{86}$ sea surface proxy based on isoprenoidal GDGTs. Since this proxy is designed for the marine environment and is based on marine archaea, it does not necessarily function in a similar fashion in freshwater systems. In lacustrine settings, the ratio between GDGT-0 and crenarchaeol (Blaga et al., 2009) is more often used to indicate the presence and potential influence of methane, as we have also done in our study. Just for your information, the Methane Index in the lake sediment core is 0.30-0.35, indeed indicating the presence of methane in Lake Loe Pool.*

Figure 3. the color coding for arable land and woodland is not easy to recognize. Please consider changing to more contrast colors.

*Reply: we have changed this.*

References:

Blaga et al., 2009. Tetraether membrane lipid distributions in water-column particulate matter and sediments: a study of 47 European lakes along a north-south transect. Journal of Paleolimnology 41, 523-540.

Weijers et al., 2006. Membrane lipids in mesophilic anaerobic bacteria thriving in peats have typical archaeal traits. Environmental Microbiology 8, 648-657.

Sinninghe Damsté et al., 2011. 13,16-Dimethyl octacosanedioic acid (*iso*-diabilic acid), a common membrane-spanning lipid of *Acidobacteria* subdivisions 1 and 3. Applied and Environmental Microbiology 77, 4147-4154.

Sinninghe Damsté et al., 2014. Ether- and ester-bound *iso*-diabolic acid and other lipids in members of *Acidobacteria* subdivision 4. Applied and Environmental Microbiology 80, 5207-5218.

Sinninghe Damsté et al., 2018. An overview of the occurrence of ether- and ester-linked *iso*-diabolic acid membrane lipids in microbial cultures of the *Acidobacteria*: Implications for brGDGT paleoproxies for temperature and pH. Organic Geochemistry 124, 63-76.

We would like to thank this reviewer for their feedback on our manuscript. Below we indicate how we will address their comments in our revised version. Our replies are in italics.

Anonymous Referee #3

Review of "Assessing branched tetraether lipids as tracers of soil organic carbon trans- port through the Carminowe catchment (southwest England)" by J. Guo. Biogeosciences Discussions, 2020.

The authors of this paper aimed to use brGDGTs as soil OC tracers in a small catchment located in southwest England and compared the concentration and distribution of these lipids in soils under different land use, riverbed and lake sediments. They showed that the relative abundance of brGDGTs does not significantly differ between soils under different land use and that brGDGTs in the riverbed and land sediments are mainly produced in situ (in the water column and/or sediment). Therefore, they cannot be used as soil OC tracers in this specific catchment. The analysis of brGDGTs and isoGDGTs along a lacustrine sediment core covering the last 100 yrs additionally showed that the distribution of these lipids (the degree of cyclisation of brGDGTs and the ratio of isoGDGT-0 vs. crenarchaeol) is roughly consistent with eutrophication changes over this period of time.

This study is of interest, as it is comprehensive and one of the few comparing extensively 5- and 6-methyl brGDGT distribution in soils under different land use, river and lake sediments. The paper is well-written and easy to read, and to my mind deserves publication in Biogeosciences after some revisions. The authors should sometimes be more moderate in their assertions and should avoid overinterpreting the data.

The following comments should help in improving the manuscript:

Line 13: Here, the authors mention the fact some tracers are required to quantify the fluxes of soil OC. Nevertheless, brGDGTs would be more qualitative than quantitative tracers. Therefore, this sentence should be modified.

*Reply: We agree with the reviewer, and we will change this.*

Lines 52-53: Here, I would directly say that brGDGTs are ubiquitous lipids, present in terrestrial and aquatic environments, and thus not necessarily specific soil tracers.

*Reply: We chose to follow a chronological order for our introduction, and thus first introduce the discovery of brGDGTs, followed by the development of brGDGT-based proxies, additional production in different aquatic environments (i.e. coastal marine area, rivers and lakes), and the implications of mixed sources for their use as proxies. We prefer to leave this as is.*

Line 54-55: This sentence should be rephrased, as only some of the brGDGT producers may belong to the phylum Acidobacteria. As brGDGTs were detected in various settings, it seems unlikely that they are produced by the same microorganisms everywhere.

*Reply: We will clarify this.*

Lines 77-93: It should be clearly mentioned somewhere that BIT index can be largely biased by in situ production of brGDGTs in aquatic settings (which was not taken into account in the initial hypothesis by Hopmans et al. 2004) and therefore should be applied with caution in coastal and lacustrine settings.

*Reply: We will emphasize this directly after introducing aquatic brGDGT production.*

Lines 82-90: These two studies are restrictive and specific. Other examples of studies dealing with brGDGT in situ production should be mentioned here (Miller et al., 2018, Climate of the Past; Loomis et al., 2014, GCA; Buckles et al., 2014; Biogeosciences etc.). Please also mention that in situ production of more cyclized but also more methylated brGDGTs is generally observed in aquatic vs. terrestrial settings.

*Reply: We agree that there are many more studies that show aquatic brGDGT production than the two that are mentioned in this comment. Please note that we already listed a large number of studies on aquatic brGDGT production in lines 77-79. Our selection includes those studies that were either first in suggesting that in situ production takes place in a certain aquatic environment, provided direct evidence for in situ production, or propose (quantitative) ways to identify the aquatic contribution. We do note, however, that in situ production in lakes is not further clarified in our manuscript. One reason for this is that there is no consistent trend among lakes that enables the identification of in situ brGDGT production, in contrast to production in rivers (more 6-methyl brGDGTs) or in coastal marine environments (higher degree of cyclisation). We will add this information to the introduction of our revised manuscript and add the appropriate references.*

Lines 100-102: In order to trace soil OC with brGDGTs, these lipids should be mainly derived from soils, with only reduced in situ production. Such an assumption should be clearly specified.

*Reply: We will add this.*

Lines 160: Were some samples analysed in replicates?

*Reply: No, we did not analyze samples in replicates.*

Lines 172-181: IsoGDGT-0 concentrations are only reported for the lacustrine sediments. What about the soils and the riverine sediments?

*Reply: We only reported the concentration of isoprenoid GDGTs for the lacustrine sediments as we only discuss them for this environment as part of the GDGT-0/crenarchaeol ratio (section 4.4). Concentration data for GDGT-0 in the other environments will be added to the supplementary table in the excel file.*

Line 227: principal component analysis instead of principle component analysis

*Reply: Thanks. We will correct this.*

Line 236: In Fig. 4b, a lot of samples are outside the circles (the 3 groups of soils) and do not overlap. This should be acknowledged.

*Reply: There are several ways to display these results. We here followed the approach of Glendell et al. (2018), who previously studied the same set of samples. The circles in Fig. 4b represent the 95% confidence interval around the mean point of the group (the enlarged symbol inside the ellipse), which is the reason why there are multiple points that plot outside the ellipse. We will clarify this in the figure caption.*

Line 251: Regarding the turnover of brGDGTs in soils, please also refer to the publication by Huguet et al. (2017, GCA), with turnover times between 8 and 41 years in the same range as Weijers et al. (2010).

*Reply: We will add this reference.*

Lines 269-270: please specify the 2 transects along which large spatial variations in BIT are observed. T1 and T2? All the discussion about spatial variations in BIT and soil moisture remains very speculative. How can you explain that these variations occur only along 2 transects? What about the other transects? Are they any in situ measurements of soil moisture available to strengthen the argumentation? Or measurements in the lab (after having dried the soil samples)?

*Reply: The BIT index values gradually increase from the hilltop downwards along Transect-1 and Transect-8. As can be seen in the table below, Transect-1 and Transect-8 show the largest change in BIT index vales (>0.3). Transect-2, Transect-3 and Transect-7 also show an increase from hilltop downslope, albeit to a smaller degree (0.17, 0.19 and 0.04 increase, respectively). The other three transects (Transect-4, Transect-5 and Transect-6) in north catchment have stable BIT values, and the BIT values in south catchment do not show an obvious trend at all. Also based on the comments of Dr. Sparkes, we will clarify our discussion on the BIT index in a revised version.*

*Unfortunately, the soil water content was not analyzed.*

Table1. BIT values along 14 transects (Tx indicates the transect number, and Sx indicates the sample point, where 1 represents the hilltop and subsequent numbers are further downslope).

| | BIT | \*T1\* | \*T2\* | \*T3\* | T4 | T5 | T6 | \*T7\* | \*T8\* | T9 | T10 | T11 | T12 | T13 | T14 |
|---|---|---|---|---|---|---|---|---|---|---|---|---|---|---|---|
| | | North catchment | | | | | | | | South catchment | | | | | |
| hilltop | S1 | **0.65** | *0.66* | *0.77* | 0.97 | 0.99 | 0.97 | *0.77* | **0.58** | 0.92 | 0.92 | 0.84 | 0.95 | 0.84 | 0.86 |
| | S2 | **0.57** | *0.66* | *0.86* | 0.99 | 0.99 | 0.93 | *0.65* | **0.59** | 0.91 | 0.63 | - | - | 0.73 | 0.88 |
| | S3 | **0.73** | *0.77* | *0.87* | 1.00 | 1.00 | 0.94 | *0.82* | **0.80** | 0.98 | 0.90 | 0.92 | 0.98 | 0.91 | 0.88 |
| | S4 | **0.88** | *0.83* | *0.96* | 0.99 | - | 0.96 | *0.70* | **0.85** | 1.00 | 0.92 | 0.72 | 0.98 | 0.92 | 0.86 |
| | S5 | **0.95** | - | - | - | - | 0.97 | *0.76* | **0.98** | 1.00 | 0.91 | 0.91 | 1.00 | 0.91 | 0.93 |
| | S6 | - | - | - | - | - | 0.96 | *0.94* | **0.97** | 1.00 | 0.93 | 0.85 | - | 0.90 | - |
| | S7 | - | - | - | - | - | - | *0.81* | - | 0.95 | - | 0.92 | - | - | - |
| downslope | S8 | - | - | - | - | - | - | - | - | - | - | 0.92 | - | - | - |

Line 277: similarly, please specify the 4 transects along which large spatial variations in IR index are observed.

*Reply: We will add the specifications.*

Lines 281: Is the relationship between the relative abundance of 6-methyl brGDGTs and pH given for all the soils of the catchments or only those of the 4 transects previously mentioned?

*Reply: The reported relationship between the relative abundance of 6-methyl brGDGTs and pH is for all the soils in the study catchment. We will specify this in the manuscript.*

Lines 281-283: similarly, please specify to which soils correspond the different pH values (those of the 4 transects, the total dataset etc).

*Reply: We will further specify this.*

Lines 321-322: In addition to Congo, brGDGTs are also mainly derived from soils in other large riverine systems such as the Amazon (Kim et al., 2012, GCA) or Rhône river (Kim et al., 2015, Frontiers in Earth Science).

*Reply: We will add these studies.*

Line 326: why would brGDGTs would be degraded more rapidly in soils than in aquatic settings? This sentence should be removed as it appears too speculative.

*Reply: The line that the reviewer refers to is on purpose phrased as a potential explanation for our results, and thus meant to be speculative Note that we do not compare brGDGT degradation in soils vs an aquatic setting, but the degradation of soil-derived vs aquatic brGDGTs in the same aquatic environment. One process that could explain this process is priming. We will add this explanation and appropriate references (e.g. Bianchi, 2011) to the revised version.*

Line 348-349: as said above, the identity of the brGDGT producers remains elusive in soils as well.

*Reply: Yes, we agree with the referee, although so far there are more clues on the producer(s) of brGDGTs in soils than there are for aquatic systems.*

Lines 352-358: I do not see the interest of this part of this discussion on the ecological niches of brGDGTs producers in Loe Pool as it is totally speculative and has no direct link with the main aim of the paper (using brGDGTs as soil OC tracers).

*Reply: In this section it becomes clear that the brGDGTs in the lake sediment are not derived from soils, but are most likely produced in the lake itself. Since we can, therefore, not use the brGDGTs as tracer for soil OC, we instead use this section to further explore the environmental significance of their signature stored in the lake sediments. For this, it is important to understand the depth and season of brGDGT production in Lake Loe Pool, for which we compare our dataset with the latest insights on brGDGT production in lakes in general, i.e. the ecological niches identified in Lake Lugano (Weber et al., 2018).*

Lines 359-389: in this section about local environmental changes, what about reconstruction of past temperature/pH variations with brGDGT-based indices? It would be complementary to the discussion about the lake eutrophication.

*Reply: We agree with the reviewer that records of past temperature and pH variations would be a valuable addition to the discussion. However, the aquatic source of the brGDGTs in the lake sediments disqualifies the use of the transfer functions from e.g. De Jonge et al., 2014 or Naafs et al., 2017, that are based on soils. We did apply the transfer functions in the latest lake calibration (Russell et al., 2018), however, the calibration dataset only includes lake sediments from tropical east Africa, and results in reconstructed temperatures that are too high (13.7 ± 0.1 °C vs the locally historical recorded temperature of 10.9 ± 0.6 °C (average of 1978 to 2018, UK metoffice)). It thus seems that both the global soil calibration and the tropical lake calibration are not appropriate for the brGDGTs in this temperate lake, and therefore decided to not include these records in our manuscript.*

Lines 385-387: The authors should also mention the in situ production of isoGDGTs in deep lacustrine sediments, as it could bias the signal recorded in the sediments.

*Reply: We will add that isoGDGTs may potentially be produced in deeper sediments, although we are not aware of a study that has shown this and we can add as a reference. Given the resemblance of the trends in GDGT proxies with that of the eutrophication history of the lake, we also assume that the contribution of a deep-sediment-producer will be minor.*

Lines 395-397: I would rephrase this sentence. There is no direct evidence that soil moisture exerts a control on brGDGT distribution here and the variations in BIT were observed along 2 transects only.

*Reply: As we mentioned above, the trend in BIT values is evident in five out of eight transects in the north catchment, although the increase is relatively small in three of them. Based on the influence of soil water content reported in the literature (e.g. Dirghangi et al., 2013; Menges et al., 2014), and the supposedly lower ground water table at the hilltop compared to the soils downslope, we will leave this interpretation as is.*

Line 401: Please replace "replaced" by "mixed", as the soil brGDGT signal is not replaced by the aquatic brGDGT signal, the two signals are mixed in the sediment.

*Reply: We will change this sentence.*

Lines 407-410: please be more moderate here, as the interpretation based on brGDGTs is purely qualitative and complementary to previous data. I would rather say that the trends derived from GDGT data are roughly consistent with the historical record of lake eutrophication.

*Reply: We will change this accordingly.*

Lines 411: this sentence should be modified, as in the case of the Carminowe Creek catchment, this study clearly showed that brGDGTs do not record land management change and that in situ production dominates in the riverine system.

*Reply: Note that we here refer to GDGTs in general, not just the brGDGTs. The land management that we mention refers to the increased use of manure and septic tanks and intensified agriculture that caused the eutrophication of the lake, and the subsequent restoration efforts that are reflected in the GDGT proxy records from the lake core (Fig. 6). The conclusion that brGDGTs in the lake sediments are produced within the lake is already clearly mentioned in line 402-403.*

We would like to thank Robert Sparkes for his kind and encouraging words. We appreciate his comments and indicate how we will address them in a revised version of our manuscript below. Our replies are in italics.

Dr. R. Sparkes

Guo and colleagues have carried out an in-depth study of GDGT mobilisation, transport and production in single catchment, Loe Pool Lake, in the UK. They attempt to use branched GDGTs as tracers for soil mobilisation from different parts of the catchment, but conclude that this is not possible using the available samples. Instead they use data from the creek and lake to reconstruct catchment-wide changes through the last century.

This is an exemplary study, demonstrating how to carry out a thorough investigation of GDGT data. The authors should be applauded for their careful application of laboratory, analytical and statistical techniques. I recommend that this paper is accepted for publication.

Minor comments are limited to typographical and grammatical changes:

Line 166: The word 'close' is repeated

*Reply: We will delete the repeated 'close'.*

Line 256: Extra space ( Weijers

*Reply: We will delete the space.*

Line 268 – 271: These sentences do not make it clear the direction of the trend. "BIT values gradually increase > 0.4" lacks context – do they increase or decrease with altitude? Also, be specific about whether crenarchaeol is decreasing up or down the transect

*Reply: The BIT index values gradually decrease with altitude, i.e. they are lowest on the hilltop. The change in BIT index values is driven by both an increase in the amount of brGDGTs and a slight decrease in the amount of crenarchaeol from the hilltop down the transect. We will specify the trend of the BIT values along different transects in our revised manuscript.*

Line 277: "Interestingly, also . . ." is grammatically odd

*Reply: We will change this to "Interestingly, the IR is also…".*

Line 315: suggest replacing 'occupied' with 'accounted for' or similar words

*Reply: We will replace this.*